

# Wind turbine wake detection and characterisation utilising blade loads and SCADA data: a generalised approach

Piotr Fojcik[1], Edward Hart[1], and Emil Hedevang[2]

[1]Electronic & Electrical Engineering, University of Strathclyde, 99 George Street, Glasgow G1 1RD, United Kingdom
[2]Siemens Gamesa Renewable Energy, Borupvej 16, 7330 Brande, Denmark

**Correspondence:** Piotr Fojcik (piotr.fojcik@strath.ac.uk)

**Abstract.** Large offshore wind farms face operational challenges due to turbine wakes, which can reduce energy yield and increase structural fatigue. These problems may be mitigated through wind farm flow control techniques, which require reliable wake detection (recognising the presence of a clear wake) and characterisation (parametric description of a wake's properties) as prerequisites. This paper presents a novel three-stage framework for generalised wake detection and characterisation. First, a regression model utilises blade loads and SCADA data to estimate the wind speed distribution across the rotor plane. Second, a Convolutional Neural Network (CNN) undertakes pattern recognition analysis to perform the wake detection, classifying rotor-plane wind estimates as "fully-impinged", "left-impinged", "right-impinged" or "not impinged." Third, where wake impingement is detected, 2D Gaussian fitting is undertaken to provide a parametric wake characterisation, providing outputs of the wake centre location and wake lateral width. The framework is tested and assessed using a virtual wind farm in the North Sea and a wide range of wind conditions (mean ambient wind speeds from 5-15 m/s, turbulence intensities from 3-9%, full range of wind directions). Results show high accuracy of wind field estimation, with the mean RMSE over all test cases being 0.351 m/s, or 5.23% when normalised by mean ambient wind speed. A wake detection sensitivity study confirms accurate performance across a majority of wind conditions, with minor issues observed only for more extreme conditions or those at the limits of the utilised training data. The final wake characterisation stage is shown to flexibly adapt to changing wind conditions, successfully tracking the wake's position even in more demanding partial-impingement cases. The proposed framework therefore demonstrates strong potential as a generalised approach to wake detection and characterisation.

## 1 Introduction

The wind industry has advanced significantly in recent years, reaching a total worldwide installed capacity of 906 GW at the dawn of 2023 (Hutchinson and Zhao, 2023). Despite being a mature field, some challenges still remain to be solved. Due to the increasing size of modern wind farms, there is a newly-posed challenge of effectively operating clusters of hundreds of multi-megawatt machines. For this reason, the current focus for both research and industry is shifting towards a farm-level approach for operations, control and maintenance.

A wind turbine interacts with the incoming wind flow, extracting energy and creating a wake: a downstream region of decreased wind speed and increased turbulence. Due to their varied impact on the farm's performance, the wind turbine wakes





are an important aspect of the aforementioned task. First of all, operating a wind turbine affected by wake deficits from the upstream machines results in a lower energy yield because of wake-generated velocity deficit in the upcoming flow (Adaramola and Krogstad, 2011; Barthelmie et al., 2010). Having the rotor experience an altered wind field also leads to uneven aerodynamic loading on the machine, inducing severe fatigue cycles to its structure (Churchfield et al., 2012). A turbine under waked flow conditions is also prone to experiencing additional turbulence due to large-scale motions of wake meandering, creating
substantial loads and thus fatigue (Madsen et al., 2010; Larsen et al., 2013).

Via various control techniques it is possible to mitigate the negatives of waked flow conditions, optimizing turbine operation for reliable and effective performance. Probably the earliest developments in this area date back to a study by Steinbuch et al. (1988), where the authors implemented axial induction control to increase the energy yield by reducing the wake effects. Since then, numerous methods for optimizing the wind farm flow have been developed; some examples include yaw control towards
directing the wake away from the downstream machine (Howland et al., 2020; SGRE, 2019), or cyclic pitch control towards inducing improved mixing in the wake (Frederik et al., 2020).

For the sake of the discussion in this work, we introduce two terms: *wake detection* and *wake characterisation*. By the former we refer to the action of recognising that a given turbine is experiencing a clear wake impingement from a nearby machine, in such a way that it has a significant (in both magnitude and time) influence on its performance. By the latter we refer to the
action of identifying the properties of the wake using a simplified parametric representation, and monitoring how the values of these parameters change in time. These two steps are key prerequisites to wind farm flow control, which is why lately there has been a noticeable effort in the research community towards developing methods to obtain detailed information on the instantaneous wind field impinging downstream turbines. So far, these approaches have focused on implementing remote sensing techniques or using a turbine's operational data. The former is usually achieved with the use of LIDAR devices - some
examples include works by Lio et al. (2020) and Raach et al. (2016). A different way of tackling the problem is to employ the relationship between wind field and turbine response (primarily blade-root bending moments), which allows for a comparable accuracy in wake detection and characterisation (Onnen et al., 2022; Dong et al., 2021; Farrell et al., 2022).

Despite achieving good wake tracking performance, the LIDAR-based approaches unavoidably rely on an additional hardware, which is not currently present at most wind farms and represents a significant additional cost. As for the load-based
methods, to-date the studies focused solely on a scenario of a direct and guaranteed wake impingement from an upstream turbine. This therefore only handles a part of the overall problem in a realistic setting, since for the majority of wind directions the turbine will not be directly affected by a nearby turbines wake, it will however receive a large amount of random flow fluctuations due to operating in the highly turbulent atmospheric boundary layer. Without knowing whether the turbine operates under wake impingement or not, there is a risk of implementing a wake characterisation scheme on a naturally occurring
turbulent eddy, hence providing incorrect and misleading inputs to the control system. For this reason, the vast majority of methods developed so far are not yet applicable to real world operations. Therefore, to the best of the authors' knowledge, there has not previously been a generalised method propoposed, which allows for both wake detection and a subsequent wake characterisation (if the flow field is recognised as being wake-impinged).





In this paper, we seek to bridge the above identified gap by proposing a generalised approach that performs both wake detection and characterisation within a single framework. Firstly, the instantaneous wind field interacting with the rotor is estimated using turbine operational and load data. Secondly, a Neural Network approach is implemented for classifying whether the estimated wind field represents a case of 'full', 'partial' or 'no' wake impingement. Thirdly, if clear wake impingement is detected, a parametric wake model is fitted in order to characterise the centre and lateral-width of the currently observed wake. A full end-to-end methodology is presented, which aims to provide both a demonstrator and performance benchmark for generalised wake detection and characterisation methods of this type. It is highlighted that focus of the current work is that of developing a solution which is able to confidently assert when a turbine is impinged by a wake from a nearby turbine, as this information is critical to farm level wake steering control.

The paper is structured as follows: Sect. 2 introduces core concepts which are key for understanding the scope of this work and provides a literature review of state-of-the-art in wind field reconstruction and wake estimation techniques; Sect. 3 discusses the methodology used for the framework's development and training; Sect. 4 presents the results of the framework validation; Sect. 5 considers the discussion on the framework applicability and limitations; and lastly, Sect. 6 covers the conclusion and addresses future work.

## 2 Background

This section describes the concepts of Atmospheric Boundary Layer, physics of turbine wake and wind farm flow as well as numerical turbine/farm flow models. It also covers the state-of-the-art in wake estimation and explains some of the methods used in the scope of this work.

### 2.1 Atmospheric Boundary Layer

The turbines operate in the atmospheric boundary layer (ABL), a highly turbulent environment where all motions are affected by the proximity of the Earth's surface. Its height, usually associated with a characteristic value of 1 km (Calaf et al., 2010), can vary depending on the weather conditions and diurnal cycle. It introduces complex gradients of heat, momentum and humidity that strongly influence the wind farm's performance (Stevens and Meneveau, 2017). The general behaviour of the ABL is often described with its stability, which is related to the vertical temperature distribution and can be one of three types: *unstable* conditions (increased vertical fluctuations, horizontal flow turbulent), *stable* conditions (reduced vertical fluctuations, horizontal flow near-laminar) and *neutral* conditions (insignificant effect of vertical movements to the horizontal flow) (Keck et al., 2014). The terms *wind shear* and *wind veer* refer to changes in wind speed and direction with increasing height, respectively. The turbulent flow structures - termed *eddies* - are present in the ABL at different time and space scales. Most relevant for this project is the microscale, covering changes occurring over seconds or minutes with the length scales of up to 1 km. The most common way to describe the severity of the atmospheric fluctuations is the parameter of ambient turbulence intensity $I_{amb} = \sigma_U / \overline{U}$, where $\sigma_U$ is the standard deviation of the stream-wise velocity fluctuations and $\overline{U}$ is the average stream-wise velocity.





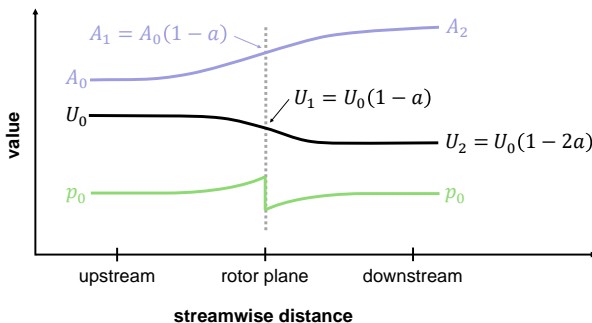

**Figure 1.** A diagram showing how the rotor area $A$, mean wind speed $U$ and mean pressure $p$ change due to the flow interacting with the rotor. Rotor plane presented with a dotted grey line.

## 2.2 Wind turbine wake physics

A significant effort was put in by the scientific community to determine how a wind turbine interacts with the incoming flow. Thanks to decades of numerical, wind tunnel and field studies (Ronsten, 1992; Whale et al., 2000; Barthelmie et al., 2004; Medici and Alfredsson, 2006; Chamorro and Porté-Agel, 2009; Hamilton et al., 2015) this topic is well understood today. The recent literature is also enriched with detailed review articles (Stevens and Meneveau, 2017; Porté-Agel et al., 2020; Meyers et al., 2022) covering numerous aspects of the wind turbine and wind farm flow.

The actuator disc concept (Froude, 1889), being the cornerstone of most analytical techniques used today, is a reasonable simplification of the flow transitions happening at the wind turbine and is thus a proper introduction into any discussion on the wake subject. It considers the rotor as an infinitely thin permeable disc, inducing an axisymmetric force on the passing flow. As the flow approaches the moving blades, it is being slowed by the resistance of the rotor. As shown in the Figure 1, the initial velocity value of $U_0$ is gradually reduced to a value of $U_0(1-a)$ at the rotor plane. The value of $a$ is referred to as the axial induction factor. To satisfy the mass flow conservation condition, decreasing velocity results in the cross-sectional area $A$ of the flow increasing. The energy extraction occurs only at the rotor plane, so before the flow reaches that point, its energy must remain unchanged. At the rotor plane, the extraction of energy manifests as a sudden drop in pressure. In the downwind region, the velocity keeps decreasing alongside a gradual increase in pressure until reaching the initial (atmospheric) level of $p_0$. The energy extraction is now visible in the velocity deficit $(U_2 - U_0)$. From the above discussion it should be clear that: a) kinetic energy is extracted from the flow at the global level (not locally); b) this is driven by the turbine extracting pressure (potential) energy at the local level.

The downstream region of the stream-wise flow is referred to as *wake*. It can be divided into the near-wake, considered to be up to 2-4 rotor diameters from the rotor, and the far-wake, spanning further downstream. These two flow structures are significantly different. The well-structured near-wake is directly linked to the rotor properties, manifested with tip and hub



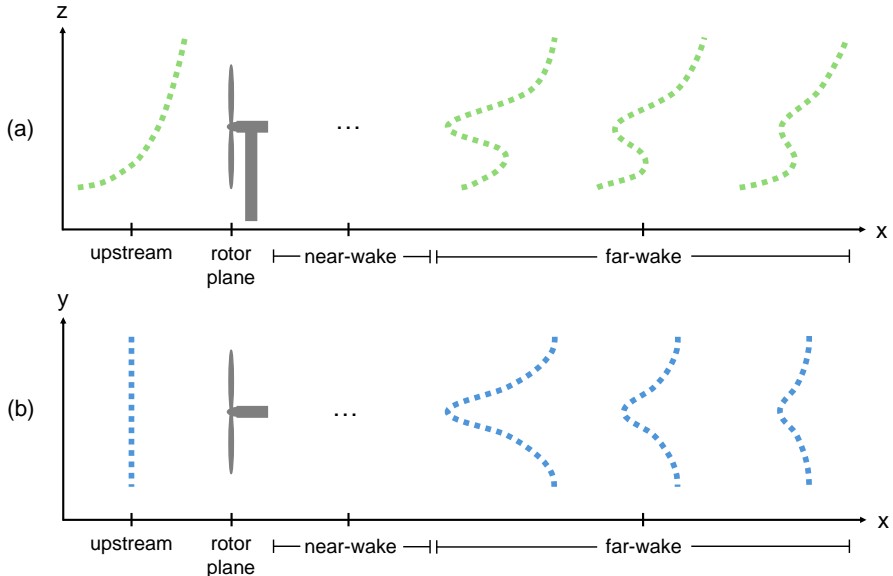

**Figure 2.** The evolution of mean wind speed profiles with the ABL flow moving in the streamwise direction $x$; shown as (a) function of vertical distance $z$ (view from the side), (b) function of horizontal distance $y$ (view from above). Near-wake profiles omitted due to not being the focus of this work. Adapted from (Stevens and Meneveau, 2017).

vortices. This region is of secondary importance for this discussion, as the modern turbines are spaced at such distances that they usually are in the far-wake of their upstream neighbours (Meyers et al., 2022). In the far-wake region, with the organised vortices dissipated, the flow is able to recover its kinetic energy by entrainment of the surrounding flow. The velocity deficit is gradually minimised, while the wake width increases. For the sake of simplicity, in this work the term 'wake' refers to the far-wake flow region.

In an idealised approach, where the wind field is simplified to a lateral flow with no ground present, mean velocity profile in a wake follows a near-perfect[1] Gaussian distribution; also, with $x$ considered to be downstream distance, the velocity deficit value at the rotor centre decreases proportionally to $x^{-2/3}$, while the wake width grows proportionally to $x^{1/3}$. The non-disturbed mean velocity profiles are characterised with self-similarity, meaning that the deficit shape (normalized by its maximum value), considered as a function of the radial distance from wake centre (normalized by wake width), remains the same along the streamwise direction.

With this baseline explained, one should consider the wake physics in a more realistic setting. Figure 2 shows the mean $U$ profiles typical for the ABL wind field and how they are altered after interacting with a rotor. Firstly focusing on the flow velocity, it can be seen that the stream-wise wake development defers from the idealised scenario mentioned previously. First of all, due to the atmospheric wind shear effect, the mean velocity profile is skewed towards the ground, resulting in a higher value in the upper part of the rotor. In (Stevens and Meneveau, 2017) the authors present this effect by comparing several different

---

[1]There are corrections for e.g. hub and blade tips.





numerical and experimental studies. Furthermore, extensive numerical (Peña and Rathmann, 2014), wind tunnel (Bastankhah and Porté-Agel, 2014) and field (Fuertes et al., 2018) studies have shown that the atmospheric turbulence causes the wake

width to grow approximately linearly with increasing downstream distance. A more rapid wake growth relates to faster energy recovery, which was observed to be the case in wind fields with higher turbulence. That is the reason why wakes persist longer in stable atmospheric conditions, where the turbulence is lower (Wu and Porté-Agel, 2012).

In terms of the turbulence itself, the wake structure is strongly shaped by both the rotor's performance and the atmospheric fluctuations. The former occurs due to blade- and hub-induced vortices, which enhance the turbulence intensity already present

due to the atmospheric fluctuations. The vertical turbulence intensity profile is affected by the atmospheric shear, which is why the peak $I_{amb}$ value is present at the top of the rotor. While the smaller-scale eddies are mostly responsible for the energy recovery, the large-scale lateral motions can significantly alter the wake's path, a phenomenon which was termed *wake meandering*. It results in downstream turbines experiencing wake effects intermittently in time, which in turn causes unsteady loads. Experimental studies (España et al., 2011) indicate that meandering seems to occurs only if the incoming flow has eddies

much larger than the rotor diameter.

## 2.3   Wind farm flow physics

The flow entering a cluster of wind turbines undergoes additional transformations compared to the ones discussed in Sect. 2.2. For the first several rows of turbines, the flow is highly heterogeneous due to the individual wakes being formed and interacting with each other. This region is referred to as *entrance and flow development region*. While moving further downstream, the

flow remains heterogenous at the turbine height, but the flow in higher regions of ABL eventually stabilise, allowing for energy recovery and thus power production to be more balanced. This situation, being only possible for very large wind farms, is referred to as *infinite wind farm case* (Porté-Agel et al., 2020).

Of prime interest for the project is the first of the above-mentioned flow regions, as the wakes there are not heavily mixed with each other yet and the application of wind farm flow control is much more feasible. The power degradation is strongly

pronounced while moving downstream through the first few rows of the wind farm. A study by Barthelmie et al. (2010) presented the impact of wind turbine wakes to the energy production in Horns Rev and Nysted offshore plants, located on the North Sea and Baltic Sea, respectively. The field measurements were done for inflow from different wind directions, analysing the power loss across 10 (Horns Rev) and 8 (Nysted) first turbine rows. The study showed a significant drop in power production - the turbines located the 'deepest' in the wind farm would only have the output of approximately 45% or 70% of the first-row-

turbine value, depending on the wind direction. Where the inflow angle would cause the turbines in a column to be directly downstream from each other, the major drop would be immediately after the first row, and remain fairly similar. In other cases, the gradually mixing wakes would result in a more gradual drop in power production. This study, along with others (Walker et al., 2016; Nygaard, 2014), proved the great importance of wake distribution to the power production of a wind power plant.





## 2.4    Numerical turbine/farm wake models

The purpose of engineering wake models, used extensively in both academia and industry, is to estimate the wake development and its effect on the downstream turbines - which then can be used to approximate the power output, machine loads and other key parameters. These approaches rely on implementing necessary simplifications to model the turbine flow with low computational cost. The Jensen model (Jensen, 1983; Katić et al., 1986), despite being developed more than 40 years ago, is still widely used in the industry and provides a basis for more advanced models. It considers a simple top-hat velocity deficit

profile based on thrust, rotor radius, downstream distance and a wake expansion coefficient, the latter dependent on aspects such as atmospheric conditions, topography etc. Newer models tend to consider the wake velocity distribution to be a Gaussian profile, which is a decision based on experimental studies of the wake - a study by Bastankhah and Porté-Agel (2014) is the one that gained most attention in that regard. A more sophisticated approach was taken by Ainslie (1988), who derived the model from the approximated Navier-Stokes equations. The model copes well with representing the wake interaction with the

ABL, which is due to consideration of turbulent eddy viscosity, analysing how the different turbulent eddies react with each other. In a recent study (Kim et al., 2018), a variant of the Ainslie model proved to be more accurate than the popular Jensen or Bastankah-Porté-Agel models in matching experimental wind tunnel data.

The Ainslie model was the first to consider both the small and large scales of the turbulence in the context of wake development. It was an inspiration for the more recent Dynamic Wake Meandering (DWM) model (Larsen et al., 2007), which is

currently the state-of-the-art for wind turbine load and power modelling, recommended in the latest IEC standard (IEC, 2019). In its essence, the DWM model considers the wake as a passive tracer being moved laterally and vertically by atmospheric motions. It is achieved by turning the wake into a cascade of wake deficits, being 'released' by the turbine with high frequency, moving with an advection speed equal to the ambient flow speed. The path of each tracer is dictated by large-scale stochastic turbulence, in the form of eddies with a characteristic length of twice the rotor diameter or larger. The defined tracer path

creates a meandering frame of reference, in which the Ainslie-based velocity deficit and turbine-induced turbulence effects are then separately captured. The result is a model which distinguishes the impact of turbulence: overall wake path dictated by large-scale eddies, and wake energy recovery ordered by small-scale ones. The model requires some form of stochastic turbulence field mimicking the atmospheric conditions, such as Kaimal model (Kaimal et al., 1972) or Mann model (Mann, 1994).

## 2.5    State-of-the-art in wind field reconstruction


Most of the recently developed inflow estimation methods which aim to capture more dynamic wind phenomena without using external equipment such as LIDARs are based on incorporating the rotor loads. The widely discussed in the literature method introduced by Bottasso and Schreiber (2018) relies on using the out-of-plane blade bending moments to approximate the local inflow wind speed at a given azimuth position. The blades act here as a tool to sample the spatially non-uniform incoming wind

field. The rotor is separated into four sectors, and the wind speed values at each are estimated from the measurements using an estimator based on Kalman filtering – this way, the vertical and horizontal shear profiles on the rotor plane can be effectively



approximated. The approach was later tested in the field (Schreiber et al., 2020) using a setup of two wind turbines and a met mast, showing an excellent agreement in the shear estimations between the two.

Another method based on implementing the blade loads, first described by Bottasso and Riboldi (2014), considers using the once-per-revolution (1P) harmonics instead of the time series. The authors proved that the vertical shear and the lateral misalignment in incoming inflow result in a specific 'trace' in the frequency domain. The estimator built on this basis was later extended (Cacciola et al., 2016), the outcome being an estimation model that maps the in-plane and out-of-plane 1P blade load harmonics to inflow conditions. In a field application, the model needs to be tuned using the measurements from a met mast and wind turbines in question (Bertelè et al., 2021). The wind field estimates are then produced from the live data using the least squares method or by implementing a Kalman filter. Initial field testing was performed with training data acquired from 10 Hz raw signals averaged into 1 min and 10 min values, showing a remarkable correlation between the 'sensed' values and these measured using classic mast-installed sensors. The authors pointed out that for more reliable results, there is a need for a more robust evaluation method than a met mast, which could be potentially solved using a LIDAR.

The legacy version of the above so-called 'harmonic observer' proved to have several limitations in its first form, with the most significant being the need for simultaneous measurements of all four 'wind states'. That would often lead to cross-contamination of data when e.g. one of the sensors is faulty. Another drawback is the way of dealing with nonlinearities between load signals and wind parameters. In a recent study (Kim et al., 2023), the authors decided to overcome these problems by applying neural networks to the wind and loads correlation model. A separate feed-forward neural network was assigned to each of the wind states, thus achieving a separation of the 'wind sensing' actions. The performance of the estimator was tested on the same field site as the legacy method (Bertelè et al., 2021), showing an overall similarly high accuracy – the networks trained with field data were able to follow the fast-changing behaviour of the wind field.

A similar approach to the legacy Bottasso & Riboldi method (Bottasso and Riboldi, 2014) was discussed in (Simley and Pao, 2016). The authors describe a hub-height wind speed and linear shear components estimator based on the Kalman filter. The turbine data channels used for constructing a state-space model which would be the input for the estimator consisted, among generator rotational azimuth angle and nacelle acceleration, of root bending moments expressed in a non-rotational frame. Tested with a 5 MW turbine numerical model under realistic turbulent inflow conditions, this solution proved to provide an accurate approximate estimation of the above-mentioned wind field properties.

In another reference (Liu et al., 2021) the authors use the blade loads in a different approach – a Subspace Predictive Repetitive Estimator is proposed to estimate the effective wind speed 'felt' by each blade along the span. It is done through the mapping process between the out-of-plane bending moments and wind speed, which can in turn be used to detect wake velocity deficits and thus, wake impingement effects. An interesting aspect of this method is the ability to capture partial wakes by measuring the wind speed change along the blade, showing high potential for implementation in inflow wake estimation. That being said, the FLORIS wake model used in the study is an overly simplified representation of an actual wind field, being a simple circular deficit with clear borders and centre; a more realistic study would need to be performed in order to decide the usefulness of this approach in real-life wind farm control.





This discussion can be summarised with a following statement: the area of wind field reconstruction has undergone a major development over the past years, paving the way for various control solutions that employ the information on the instantaneous and time-average state of the wind flow at the turbine. One of such applications is estimation of the wake impingement conditions, which is discussed in the next section.

## 2.6 State-of-the-art in wake detection and characterisation

So far, the methods that attempted to obtain detailed information on incoming flow, specifically focused on detecting or characterising wakes, were divided into two categories: LIDAR-based and load-based.

Light Detection and Ranging (LIDAR) devices can provide an accurate representation of the wind farm flow due to their ability to scan the wind field at high frequency. Some examples describing their use for wake properties analysis include the works by Bingöl et al. (2010), Trujillo et al. (2011) and Conti et al. (2020). In these studies, the authors managed to successfully capture such effects as velocity deficit, added turbulence and meandering, proving the usefulness of this technology for estimating detailed wake properties. For this reason, the application of a nacelle-mounted LIDAR in dynamic wake characterisation was recently investigated, using both an upwind-facing device (Lio et al., 2020) as well as with a downwind-facing setup (Raach et al., 2016). In both these studies, the measurements were used in combination with a numerical wake model, ultimately giving an instantaneous wake center position estimate which in turn could be used for closed-loop flow control.

A recent branch of research tackles wake detection and characterisation using information obtained purely from the turbine operational data. The basic idea behind this method is that a variation in the incoming wind field can be identified with changes in the turbine response, such as blade root bending moments and generator speed. The previously discussed in Sect. 2.5 approach of dividing the rotor into four quadrants and estimating the respective wind speeds was specifically tested for wake characterisation (Bottasso and Schreiber, 2018; Schreiber et al., 2016) with the idea of using the local wind speed estimates to continuously optimise an engineering wake model. Its performance was shown to be limited by several problems, most potentially due to the rather simplistic nature of the estimation in this method. The works of Onnen et al. (2022) and Dong et al. (2021) are great examples for employing the out-of-plane blade loads in a model based on the extended Kalman filter estimation, which effectively allows for dynamic wake center position tracking in both lateral and vertical directions. In another study by Farrell et al. (2022), the authors implemented a recurrent neural network trained with experimental and simulation data to estimate the lateral position of the wake centre. These models have proved to offer high dynamic wake characterisation accuracy comparable to LIDAR-based methods, while remaining highly sensitive to higher turbulence intensity values, as well as varying veer and shear conditions.

Having presented the state-of-the-art in the field, we now restate the research gap: the majority of up-to-date approaches focused primarily on wake characterisation, with a key assumption that a turbine is continuously impinged by a wake deficit. The central part of the generalised method we propose in this work is a wake detector aiming to 'sieve out' the wake-impinged flow from the incoming wind field, and thus applying wake characterisation in a much better informed manner.




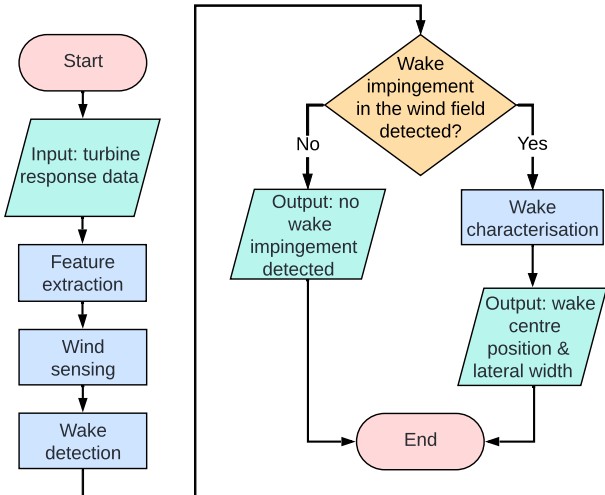

**Figure 3.** A flowchart describing the proposed wake estimation framework.

## 3 Methodology

Section 3.1 discusses the layout of the developed framework and its high-level characteristics. The details of the implementation
260 are described in Sect. 3.2 to 3.5. Lastly, the specific methods used for testing the wake sensing performance are described in
Sect. 3.6.

### 3.1 A summary of the proposed framework

Figure 3 presents the proposed wake estimation framework in the form of a flowchart. The framework consists of several pro-
cesses taking part in implementing the input turbine response data towards obtaining the ultimate information on the estimated
265 wake impingement conditions. Figure 4 presents two examples that illustrate the framework's performance when (a) the wind
field doesn't have a clear wake impingement from a nearby turbine, (b) when it does. The high-level details of the individual
models that the framework consists of are discussed below.

The first constituent model in the developed framework is a wind field estimator, which when provided with turbine response
time series as the input, is capable of producing the wind field representation, thus showing the flow at the rotor as the out-
270 put. This process is hereafter referred to as *wind sensing*. A successfully generated wind field estimation provides sufficient
information to analyze the flow field and extract the desired wake impingement information. For this framework, we propose
to perform wake detection using a convolutional neural network (CNN), which, after being trained, can distinguish the flow
with a well-defined wake deficit from a nearby turbine. If a likely wake impingement is detected, a suitable algorithm aiming
to estimate the wake's properties is to be implemented. The ultimate framework output data are two time series of the wake
275 centre position and wake lateral width, describing the horizontal distance where the degree of wake impingement is significant.





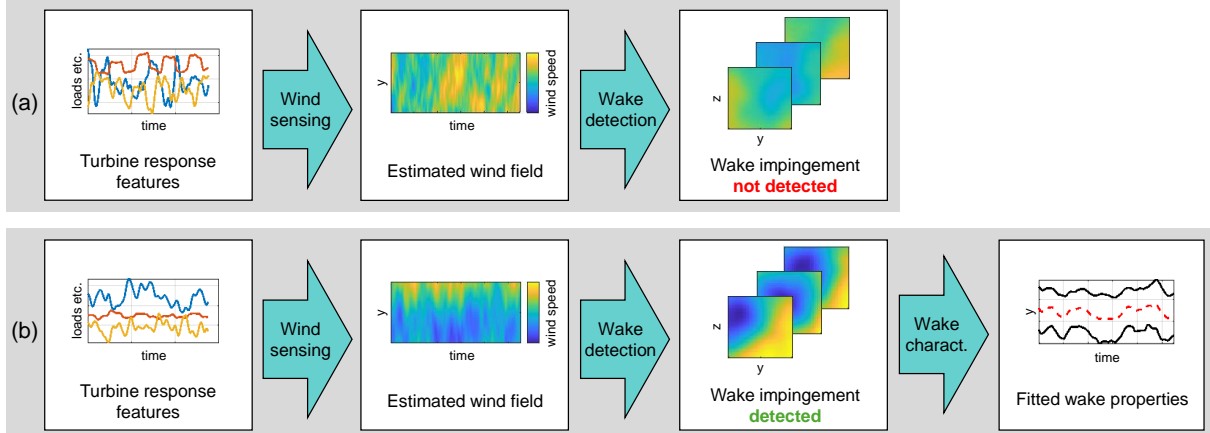

**Figure 4.** Two examples showing the framework's performance; (a) the incoming wind field doesn't have a clear wake deficit, thus the framework doesn't go into the wake characterisation; (b) there is a clear wake deficit, thus the final step of wake characterisation is implemented.

The combination of these two time series is a simplified, but qualitatively sufficient 2D representation of a meandering wake deficit. The developed methodology also allows to extract a time series of peak wake deficit $\Delta U$ value, however this is is not discussed in this work. This representation could be used as an input to the wind farm flow control techniques which are informed by the lateral properties of the impinging wake, such as the ones using yaw as the control action (Howland et al., 2020; SGRE, 2019). Focusing specifically on the lateral properties is also motivated by the fact that the wake meandering is more pronounced in the transverse direction than in the vertical direction, with the ratio reflecting the turbulence intensities of the respective components (Yang and Sotiropoulos, 2019).

## 3.2  Framework implementation: training data acquisition

A dataset comprised of 1200 simulations was produced, each one constituting of generating a unique wind field and calculating the corresponding time series of instantaneous turbine response. Synthetic wind fields were acquired by firstly generating ambient turbulence boxes using the Mann spectral tensor model (Mann, 1994) and then superimposing them with wakes generated with the SGRE's in-house implementation of the DWM model (Larsen et al., 2007). This version of the model follows the parametrisation suggested in the latest IEC standard (IEC, 2019). The turbine operation for the simulated conditions was then calculated using the aeroelastic code BHawC.

The simulations were performed for a virtual wind farm located in the North Sea. In order to train the wake detection model, turbine location and wind direction combinations were chosen such that test cases clearly differentiated between the effects of wake impingement and standalone atmospheric turbulence. With that in mind, two neighbouring turbines of interest, located near the northern edge of the wind farm, were selected: a first-row turbine 'emitting' the wake and a second-row turbine 'receiving' the wake. It was crucial for these turbines to be at the edge of a wind farm to ensure that only one wake is impacting the 'receiver' turbine; with more wakes having a chance to superimpose with each other, the impingement effects would be





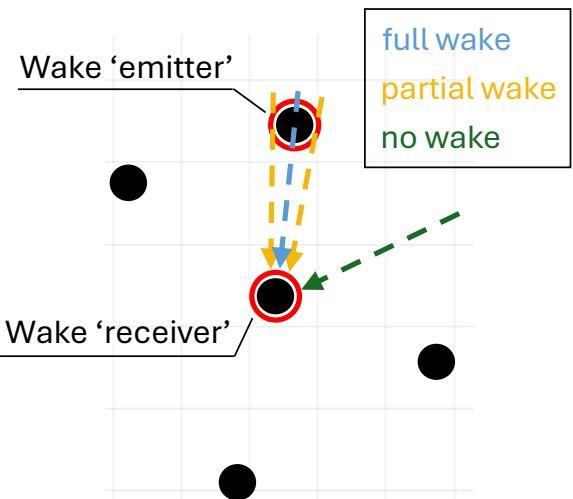

**Figure 5.** Simulated wind farm with highlighted locations of two turbines of interest and selected wind directions for the training process.

more complex and the training of a wake detection model would become more ambiguous. Four wind directions, representing four distinctive wake impingement scenarios, were defined as follows: a) fully impinged, turbine directly downstream from the closest neighbour; b) partially impinged, wake impacting the left side of the rotor; c) partially impinged, wake impacting the right side of the rotor; d) no impingement. The 'receiver' turbine would provide the turbine response data for training. The example wind farm layout as well as wind directions defining the four classes have been shown in Fig. 5.

The ambient inflow wind speed $U_{amb}$ was set as a range of 100 equally distributed values between 5 m/s and 25 m/s; ambient turbulence intensity $I_{amb}$ was set at three values of 3%, 5% and 7%. This was done to ensure that the simulated conditions would represent a multitude of different offshore wind states. Due to the fact that ambient wind fields were acquired using the Mann turbulence model, the atmospheric conditions were assumed to be neutral (Mann, 1994). The presence of vertical wind shear was simulated using the wind profile power law, with shear exponent value of 0.07 selected for all cases. It should be also mentioned that the turbines were simulated with no yaw angle misalignment.

### 3.3 Framework implementation: wind sensing

#### 3.3.1 Wind sensing input and output data selection

Figure 6 illustrates the process for training the wind sensing model and demonstrates its post-training performance in producing wind field estimations. By providing a varied dataset of turbine response time series and the corresponding wind field representations, the estimator 'learns' to approximate the wind speed distribution at the rotor from the turbine sensor data.





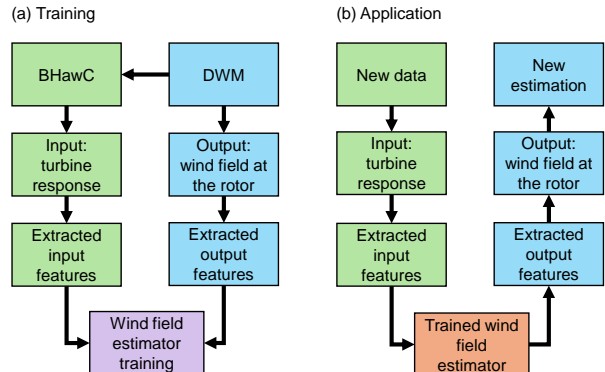

**Figure 6.** Simplified diagram visualising the flow of data during (a) training, (b) application of the wind sensing model.

The capabilities of the aeroelastic code allow to get access to an extensive set of turbine response signals like acceleration of different turbine components, generator parameters etc. To determine which data channels are best coupled with the variations in the flow, a conditional dependence was undertaken. Readers interested in the specific method are referred to (Azadkia and Chatterjee, 2019). As a result, following turbine response data was used as the wind sensing inputs: blade root bending moments in both the flap-wise and edge-wise direction; pitch angles; generator rotational speed (rpm); azimuth angle (used only for transformations of other inputs).

The wind field is considered to be a three-dimensional Cartesian grid, with three axes: X representing streamwise distance, Y representing lateral distance, and Z representing vertical distance. To represent the wind speed distribution in the flow, each point on that grid stores the local values of three wind speed components $U$, $V$ and $W$ that correspond to the spatial variation in the X, Y and Z axis, respectively. For this specific application, the approach taken was to consider the wind field as the spatial distribution of the longitudinal component $U$, as it is the one that is primarily impacted by wake deficit (Dimitrov et al., 2017). As a result, the wind fields used in the wind sensing are essentially a time series of YZ slices showing the chronological sequence of U distributions at the rotor plane.

### 3.3.2 Wind sensing input processing

Selected turbine response time series (blade root bending moments, mean pitch and rpm) were then processed to extract the optimal features from the raw data. The processes described in this section were applied both during training and the performance testing - they are an integral part of the data 'pipeline'. First of all, using the pitch signals, the edge- and flap-wise blade root bending moments were transformed to in- and out-of-plane coordinates. Performing the rotation between the two frames was calculated for a given blade $j$ as follows:

$$\begin{bmatrix} M_{j,out} \\ M_{j,in} \end{bmatrix} = \begin{bmatrix} cos(\beta_j) & sin(\beta_j) \\ -sin(\beta_j) & cos(\beta_j) \end{bmatrix} \begin{bmatrix} M_{j,flap} \\ M_{j,edge} \end{bmatrix} \tag{1}$$

where $\beta_j$ stands for the instantaneous pitch angle for a given blade indexed $j$.



Next, using the azimuth angle time series, the Coleman transformation (Coleman, 1943) was applied to the blade load data. It allowed to effectively map the separate, rotary-frame bending moments from each blade onto a stationary frame of reference.

As a result, the dynamic response to the variation in the wind can be expressed with six variables describing the loads for an entire rotor: $M_{thrust}$, $M_{tilt}$, $M_{yaw}$ (for out-of-rotor-plane movement), $M_{torque}$, $M_{lateral}$, $M_{vertical}$ (for in-rotor-plane movement). These are henceforth referred to as *rotor loads*. The Coleman transformation is defined as follows:

$$C(\psi) = \begin{bmatrix} 1/3 & 0 & 0 \\ 0 & 2/3 & 0 \\ 0 & 0 & 2/3 \end{bmatrix} \begin{bmatrix} 1 & 1 & 1 \\ cos(\psi) & cos(\psi + \frac{2\pi}{3}) & cos(\psi + \frac{4\pi}{3}) \\ sin(\psi) & sin(\psi + \frac{2\pi}{3}) & sin(\psi + \frac{4\pi}{3}) \end{bmatrix} \qquad (2)$$

where $\psi$ is instantaneous azimuth angle. It is used to transform the blade loads as follows:

$$\begin{bmatrix} M_{thrust} \\ M_{tilt} \\ M_{yaw} \end{bmatrix} = C(\psi) * \begin{bmatrix} M_{1,out} \\ M_{2,out} \\ M_{3,out} \end{bmatrix} \qquad (3)$$

$$\begin{bmatrix} M_{torque} \\ M_{lateral} \\ M_{vertical} \end{bmatrix} = C(\psi) * \begin{bmatrix} M_{1,in} \\ M_{2,in} \\ M_{3,in} \end{bmatrix} \qquad (4)$$

where $M_{1-3,out}$ is out-of-plane bending moment, and $M_{1-3,in}$ is in-plane bending moment for blades indexed 1-3.

The rotor load time series are then decomposed into its key frequency components. This process is visualised on Fig. 7.

The five frequency components selected via a conditional dependence (Azadkia and Chatterjee, 2019) study are $cos(0P)$ for describing the mean signal value, $cos(1P)$ and $sin(1P)$ for describing the once-per-revolution variation, and finally $cos(3P)$ and $sin(3P)$ for describing the thrice-per-revolution variation. A sliding window function with a width of three revolutions was applied, projecting the original load time series onto each of the the frequency components. The Fourier coefficients $\underline{a_1}$ - $\underline{a_5}$ vectors obtained from these projections are then stored, producing a coefficient time-series for each component which

eventually become the data used by the wind sensing model.

In order to capture the temporal dependencies and patterns in time series of all wind sensing inputs (including pitch and rpm), the features were embedded with their lagged values. For each sample in a time series, two additional features expressing the past value of the curve were added. These lagged features were obtained by shifting the time series by 4 and 8 seconds from the current time stamp. These specific lag values were determined by testing the framework's performance with few different

configurations and choosing the one that produced best overall results.

Due to implementing the transformations discussed in this section, using the turbine response data as wind sensing input effectively stops being a time series modelling task; instead, it becomes a time-independent regression task. Considering the entire feature extraction process, at each point in time, a 96-dimensional input vector captures and encodes the turbine's current operating state.





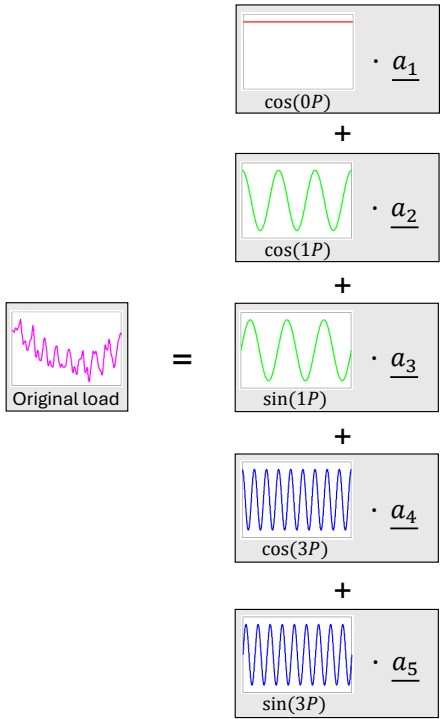

**Figure 7.** An example of load time-series decompositions into frequency component coefficient vectors ($\underline{a_1}$ - $\underline{a_5}$) capturing the contributions of each fourier mode at each point in time.

### 3.3.3 Wind sensing output processing

The three-dimensional wind field representations are being processed for the sake of data compression and efficient extraction of its key features for the best application in wind sensing. As mentioned before, the wind field is represented with a sequence of YZ slices showing the U distribution at each point along the X axis. The spatial features of each slice are then extracted, capturing the key characteristics of the wind field in a more compact form. The feature extraction is performed using the two-dimensional discrete cosine transform (DCT) (Ahmed et al., 1974). The result is similar to applying the principal component analysis; however, as opposed to this far more complex method, the DCT is a fixed transform and doesn't require any training. The DCT effectively converts the wind slice into a collection of cosine functions oscillating at different frequencies in both the vertical and horizontal directions. Each DCT coefficient corresponds to a specific frequency in the transformed domain and describes the contribution of each frequency in the distribution of U on the given YZ slice. This concept, applied to an example wind field YZ slice, is presented on Fig. 8.

The lower-order frequencies primarily capture the large-scale structures of the field, while the higher-order are more associated with encoding smaller-scale, random fluctuations. It can be seen that the most significant values can be found in the upper left corner of the diagram, meaning the majority of flow distribution is being kept by the lower orders of DCT coefficients.





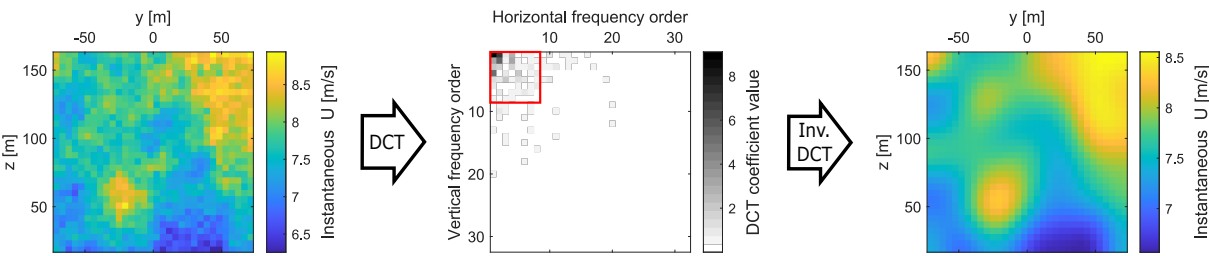

**Figure 8.** An example showing how the YZ slice of wind field (left) is transformed into a matrix of DCT components (centre) and then back into a truncated slice of wind through an inverse DCT (right). The red square indicates which DCT coefficients are stored for the implemented wind field representation.

For this reason, the key information on the flow can be easily extracted by storing only the lower orders and disregarding the
higher order coefficients. Truncating the collection of DCT order coefficients essentially acts as a spatial low-pass filter. In this work, retention of 8 DCT coefficients was found to enhance computational efficiencies while allowing for good wind sensing accuracy. The right plot on Fig. 8 shows on an example the effect of truncating the higher frequencies; it can be noticed how the wind slice is significantly 'smoothed', leaving only the major $U$ fluctuations.

### 3.3.4 Training the estimator

Having extracted the features from both the inputs and outputs, two thirds of data was then used for the wind field estimator training, while the remaining 1/3 was put aside for generating wind field estimations for wake detector testing. The current implementation utilises a localised linear regression approach, where a collection of simple models is fitted to subsets of input data, building functions that correlate the wind sensing inputs to outputs only at a narrow section of data distribution. This allowed to avoid fitting of a global non-linear function that would singlehandedly need to describe the complex relationship
between wind and turbine signals. The data points are assigned to training of a specific local linear regression model based on their pitch and rpm value, these being the key variables indicating the turbine's operating point.

### 3.4 Framework implementation: wake detection

Considering that wake detection can be essentially brought down to a classification task, we propose to deal with it by implementing a Convolutional Neural Network. The overview of this deep learning technique can be found in Appendix A. The
approach taken was to operate on the YZ snapshot samples (so-called 'front view'), showing the estimated instantaneous wind component U across the whole rotor plane, which made it possible to easily differentiate between impinged and non-impinged conditions. To capture the different forms of wake impingement accurately, a total of four classes were defined: a) fully impinged, turbine directly downstream from the closest neighbour; b) partially impinged left, wake impacting the left side of the rotor; c) partially impinged right, wake impacting the right side of the rotor; d) no detectable impingement. Figure 9 shows
selected YZ samples being appropriate examples of each one of the four classes.





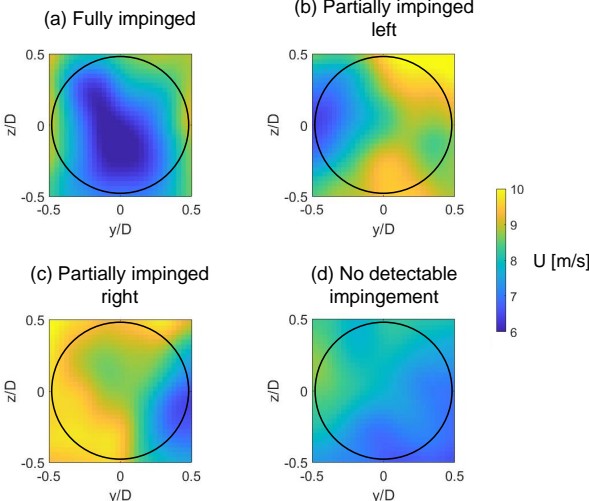

**Figure 9.** Example front-view YZ snapshot samples representing four classes defined for the neural network training. The rotor area indicated with a black circle.

The turbine response time series from the remaining 400 simulations, set aside during the wind sensing estimator training, were now used for generating wind field estimations, allowing to build a new dataset for the CNN training. An extensive manual review of these wind fields allowed for a following conclusion: in the available dataset, a clear, non-dissipated wake is only visible for lower values of the selected ambient wind speed range - typically up to 14-16 m/s, depending on the

ambient turbulence intensity. The aim of the detector is to recognize (as robustly as possible) whether the turbine is most likely experiencing a wake impingement from a nearby machine, which is why it is crucial that the training data is devoid of samples that would be arguable regarding which class they should be part of. With that in mind, the CNN was trained with data from simulations where the mean ambient wind speed $U_{amb}$ is between 5 and 15 m/s, while maintaining the full range of turbulence intensity values (3%, 5% and 7%). The YZ 'snapshot' samples were taken every 10 seconds from the available training data,

resulting in 11,200 labelled samples.

The training was conducted using Matlab's Deep Learning Toolbox. The implemented CNN architecture was based on three convolution layers and two max pooling layers. A batch normalisation layer was added after each of the convolution layers for the sake of normalisation of weight gradients and neuron activations, thus speeding up the training process. The activation function was selected as the Rectified Linear Unit (ReLU). The 90% of the available labelled data (10,080 samples) was used

for training. The samples were shuffled before the training process, as well as after each one of the four epochs. The training was repeated several times to make sure there are no deviations. The remaining 1,120 samples were used for an integrated testing cycle, which automatically occurred after the training was complete. The average accuracy of the trained network in recognizing a wake impingement case turned out to be approximately 91%.





### 3.5 Framework implementation: Wake characterisation

The final part of the developed framework is the identification of the key wake parameters in cases where a clear wake impingement is detected. This was undertaken by scanning for the characteristic velocity deficit shape in the appropriate wind field representations.

#### 3.5.1 2D Gaussian fitting

The 'tracking' of wake position and width was achieved by implementing a 2D Gaussian fitting scheme based on a least
squares algorithm. This method is a commonly used solution in the literature for the dynamic wake properties analysis (Abkar and Porté-Agel, 2015; Trujillo et al., 2011; Conti et al., 2020). Analogically to wake detection, the fitting algorithm was applied to a time series of YZ slices representing the $U$ distribution at the rotor plane, performing the non-linear least square fit based on functionalities from Matlab's Optimization Toolbox.

A general, bivariate 2D Gaussian function for spatial variables of $y_i$ and $z_i$ is expressed as:

$$f_{G,bivariate}(y_i, z_i) = \frac{A}{2\pi\sigma_y\sigma_z\sqrt{1-\rho^2}} \exp\left[-\frac{1}{2(1-\rho^2)}\left(\frac{(y_i-y_c)^2}{\sigma_y^2} - \frac{2\rho(y_i-y_c)(z_i-z_c)}{\sigma_y\sigma_z} + \frac{(z_i-z_c)^2}{\sigma_z^2}\right)\right] \quad (5)$$

where $A$ is amplitude equal to height of the peak; $\sigma_y$ and $\sigma_z$ are standard variations along Y and Z axis, respectively; $\rho$ is correlation coefficient between function's spread in Y and Z; $y_c$ and $z_c$ are means along Y and Z, respectively.

In the context of fitting the function to the velocity wake deficit, respective Gaussian parameters can be used to acquire wake properties: the function peak at $(y_c, z_c)$ can be interpreted as position of wake centre; $\sigma_y$ and $\sigma_z$ describe the spread of
wake along Y and Z axis, respectively; $\rho$ can be used to calculate the rotation angle as well as the lengths of semi-minor and semi-major axis of the wake ellipse. A Levenberg-Marquadt non-linear least-squares algorithm was employed to optimize the parameter set $(A, \sigma_y, \sigma_z, \rho, y_c, z_c)$ towards a best fit with a snapshot data. The initial guess for a function peak at $(y_c, z_c)$ was defined as the minimum U value location. The initial guess for standard deviations $\sigma_y$ and $\sigma_z$ was defined as 0.5D in order to best reflect the typical width expected of a wake.

As explained earlier, the framework outputs refer solely to the lateral wake characteristics, which are expressed by the variation on the Y axis. Out of all fitted parameters, only two then required to describe the wake variation in the lateral direction: $y_c$ and $\sigma_y$. To express the size of the wake deficit parametrically, an assumption is made that the lateral half-width is equal to standard deviation in the Y direction. With that in mind, the wake extends laterally for $2\sigma_y$, with the fitted centre $y_c$ in the middle. Figure 10 shows an example of the bivariate 2D Gaussian fit applied on an YZ wind field sample, with the key
characterised properties indicated.

#### 3.5.2 Filtering with moving average window

The lateral wake half-width and centre position time series were filtered using a moving average filtering scheme to smooth fluctuations originating from combining the independent results from each snapshot. It was done in way that would distinguish the slower variation being a result of wake meandering and attenuate the high-frequency peaks. This way, the ultimate

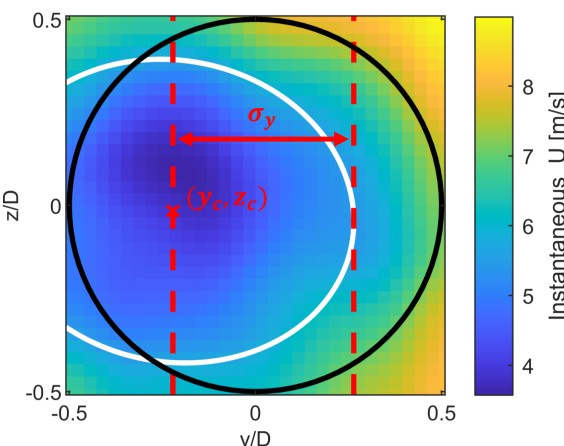

**Figure 10.** An example of a bivariate 2D Gaussian fit performance applied to a YZ wind field sample. Rotor outline marked with black line; fitted wake ellipse marked with white line; fitted wake centre at $(y_c, z_c)$ marked with red x sign; the extent of fitted lateral half-width $\sigma_y$ marked with dashed lines.

characterised wake properties time series $y_c^{filt}$ and $\sigma_c^{filt}$ were defined as follows:

$$\begin{bmatrix} y_c^{filt}(t) \\ \sigma_y^{filt}(t) \end{bmatrix} = \frac{1}{\tau} \int\limits_{t-\tau}^{t} \begin{bmatrix} y_c(t) \\ \sigma_y(t) \end{bmatrix} \tag{6}$$

The variable $\tau$ stands for window size in seconds which is calculated separately for every wind field in a following manner:

$$\tau = \frac{L_{meand}}{\overline{U_{hh}}} \tag{7}$$

where $L_{meand}$ is the characteristic meandering length equal to two rotor diameters (value corresponding to findings from the literature, used in several wake models like the DWM itself (Larsen et al., 2007)) and $\overline{U_{hh}}$ is the mean $U$ value at the hub height.

It should be noted that in this method of moving average filtering, the averaging window is not centred at the current sample; instead, the window extends from the past sample occurring $\tau$ seconds ago to the current sample. As a result, the filtered wake centre and lateral half-width time series experience a small lag of $\tau/2$.

## 3.6 Performance testing

For the purpose of testing the performance of the developed framework, several sets of simulations showing its performance for all wind directions were conducted. Each simulation set generated 360 unique wind fields, each paired with the corresponding turbine response, one for each wind direction angle. The $U_{amb}$ and $I_{amb}$ values were varied across the simulation sets, with ranges of 5, 7, 9, 11, 13, 15 m/s for the former and 3%, 5%, 7% and 9% for the latter. All simulations were performed for one





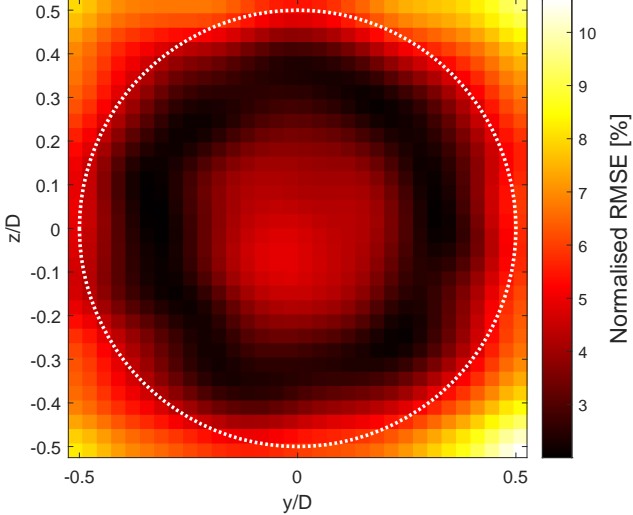

**Figure 11.** A distribution of RMSE normalised by the corresponding $U_{amb}$ value.

'receiver' turbine, which was more centrally located compared to the machine used for the training; by doing so, it experiences different types of impingement for different wind directions. Taking all of these into consideration, the performance testing dataset consisted of 3600 simulations. It was used to undertake the performance analysis of each of the three constituent models, the results of which are discussed in Sect. 4.

# 4 Results

Section 4.1 presents results of the wind sensing procedure. Section 4.2 shows the results of the sensitivity study evaluating wake detection performance as relative to the severity of ambient wind conditions. Lastly, in Section 4.3 the detailed performance of wake detection and wake characterisation together is analysed.

## 4.1 Wind sensing

The accuracy of wind field estimations was first determined by calculating the root mean square error (RMSE) between the
estimated and simulated wind speed values. For each simulation, the RMSE value was calculated separately for each location on the YZ snapshot. Figure 11 shows the YZ-plane distribution of the normalised RMSE for an example simulation. The ring-like pattern of area with lowest error is repeated across all ambient wind conditions, signifying the wind field reconstruction quality is highest at the radial distance of approx. half of the rotor radius. The flow impacting the middle blade region has the greatest impact on the root bending moments; as a result, the estimation model will be most accurate in this area. The YZ
locations outside of the rotor plane yield the highest RMSE due to blade-sensors being unable to meaningfully react to the flow fluctuations from that region.



**Table 1.** A comparison of averaged RMSE in wind speed estimation with varying $U_{amb}$ and $I_{amb}$. Last column shows values normalised by the corresponding $U_{amb}$ value.

| $U_{amb}$ [m/s] | $I_{amb}$ [%] | Av. RMSE [m/s] | Av. norm. RMSE [%] |
|---|---|---|---|
| 5 | 5 | 0.34 | 6.74 |
| 7 | 5 | 0.27 | 3.86 |
| 9 | 5 | 0.30 | 3.30 |
| 11 | 5 | 0.34 | 3.09 |
| 13 | 5 | 0.35 | 2.69 |
| 15 | 5 | 0.34 | 2.28 |
| 10 | 3 | 0.22 | 2.22 |
| 10 | 5 | 0.31 | 3.12 |
| 10 | 7 | 0.43 | 4.26 |
| 10 | 9 | 0.56 | 5.57 |

The YZ distributions were averaged across the entire rotor plane to obtain a single simulation-specific mean RMSE value. Furthermore, to check how the RMSE changes with wind conditions, mean values averaging all 360 simulations with the same $U_{amb}$ and $I_{amb}$ were calculated. Table 1 shows a comparison of mean RMSE depending on the varying $U_{amb}$ and $I_{amb}$. It can be noticed how the vast majority of cases has a low mean RMSE with the normalised value below 5% of the simulation-specific $U_{amb}$ value, indicating a very good overall wind sensing performance. Increasing the ambient wind speed results in decreasing the normalised RMSE, with the $U_{amb}$ = 5 m/s case showing significantly higher errors than others. On the other hand, the normalised RMSE was found to increase with $I_{amb}$, with the most turbulent case of $I_{amb}$ = 9% recording the largest normalised mean value of approximately 5.6%.

Figures 12 and 13 showcase example wind sensing outputs: the former presents two examples of wind fields with well-pronounced wake impingement, while the latter presents two examples of wind fields with no clear impingement.

Looking first at Fig. 12, it can be seen that the major flow fluctuations are well captured in the estimated wind field representations. Some level of smoothing can be seen to occur for the smaller-scale fluctuations in the flow, but this does not pose a major problem in the context of wake detection. In fact, due to the attenuation of less significant turbulent eddies, the wind sensing procedure acts as a spatial filter, allowing to focus only the bigger flow structures like a wake deficit. This aspect can be noticed in comparing the wind fields shown at Fig. 12: (a) portrays high $I_{amb}$, while (b) has low $I_{amb}$. It is apparent that while the some ambient turbulence structure has been smoothed in (a), the wake deficit in both cases is well captured. The wind field shown at Fig. 12 (a) is also noteworthy due to the fact that its ambient conditions of $I_{amb}$ = 9% are outside of the training dataset; however, as seen in the plot, it appears not to be a problem for the developed model, as the estimation





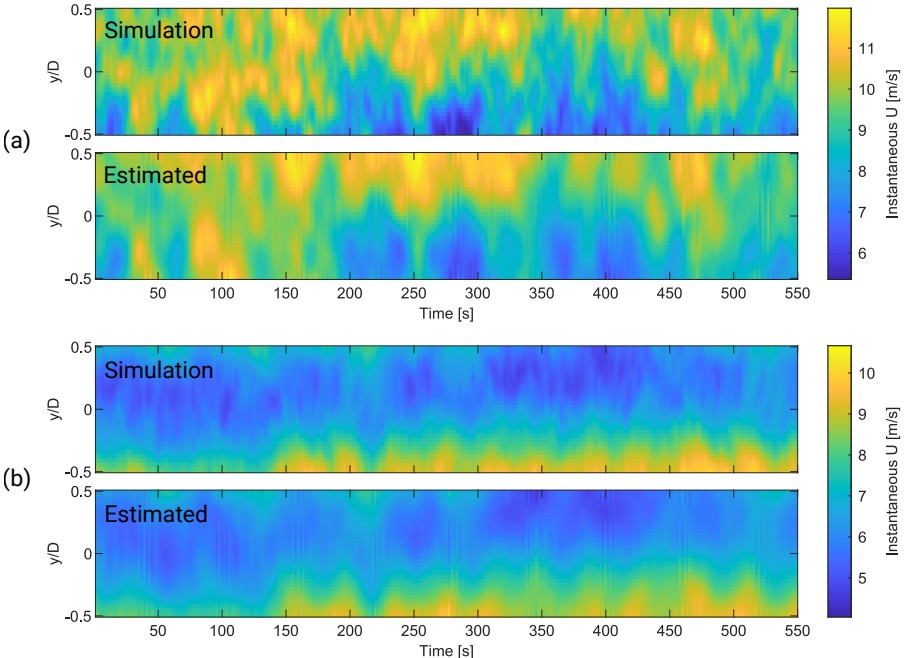

**Figure 12.** A comparison of the simulation wind fields and the estimated equivalents obtained with wind sensing. The wind fields are here presented as a time series of U distribution at the rotor plane at hub height, viewed from above. (a) Partial impingement, $U_{amb}$ = 10 m/s, $I_{amb}$ = 9%; (b) Full impingement, $U_{amb}$ = 10 m/s, $I_{amb}$ = 3%.

accuracy is still high. Overall, these results indicate that this accuracy of wind field estimation is sufficient for wake detection and characterisation.

Figure 13 shows wind sensing results for the wind fields generated with no turbines directly upstream from the 'receiver' machine. Because of that, no wake impingement should be visible at the time history of $U$ distributions, just the atmospheric fluctuations. The low $U_{amb}$ case shown in (a) reveals a slight discrepancy between the 'Simulation' and 'Estimated' subplots.
Between 0 and 300 seconds, the 'Estimated' plot displays a $U$ deficit in the central region, an effect that is not as prominent in the 'Simulation' plot. A slight disagreement would not be normally problematic; however, in this case where the flow has little overall turbulence, a subtle deficit like this can't 'hide' among other eddies, which could potentially result in classifying the mentioned samples as being wake impinged. The analysis of other low $U_{amb}$ simulations allowed to determine that this is not an isolated example; this phenomenon appears to drive the slightly higher RMSE at low wind speeds recorded in Table 1.
Looking at the high $U_{amb}$ case at Fig. 13 (b), it is apparent that as the amount of turbulence increases drastically, effectively dispersing spatially larger fluctuations that could be mistakenly classified as a wake impingement. Good overall performance is therefore seen across the vast range of tested cases, with more significant deviations generally only seen at wind speeds below 7m/s.





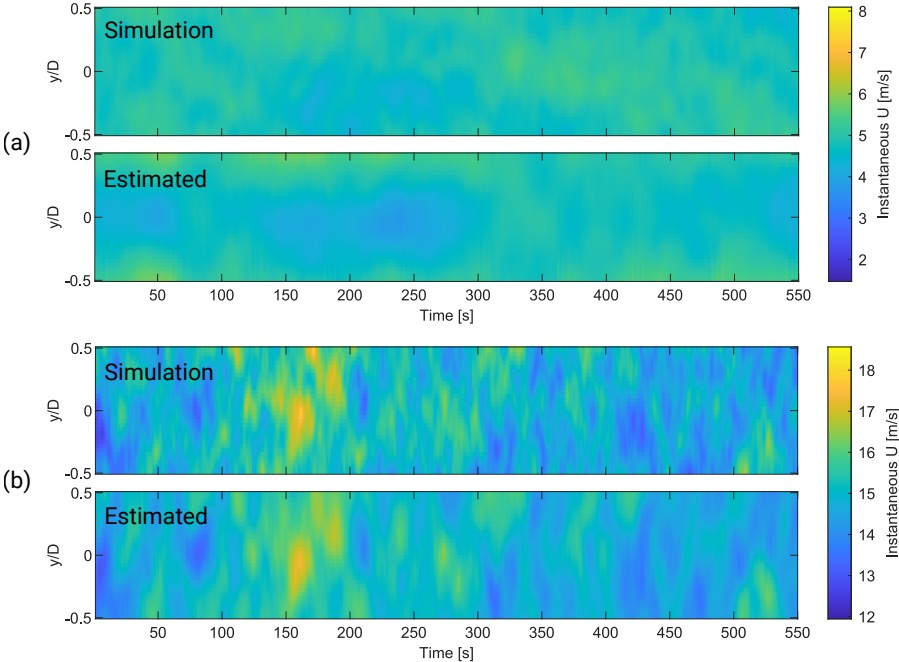

**Figure 13.** A comparison of the simulation wind fields and the estimated equivalents obtained with wind sensing. The wind fields are here presented as a time series of U distribution at the rotor plane at hub height, viewed from above. (a) No detectable impingement, $U_{amb} = 5$ m/s, $I_{amb} = 5\%$; (b) No detectable impingement, $U_{amb} = 15$ m/s, $I_{amb} = 5\%$.

## 4.2 Wake detection: sensitivity study

Figures 14 and 15 show the performance of the CNN-based wake detection framework in a full wind direction range, generated as described in Section 3.6. The figures show the sensitivity analysis for varying $U_{amb}$ and $I_{amb}$ in the flow conditions, respectively. For each $U_{amb}$ and $I_{amb}$ value, the plot displays the wind farm layout with the wake 'receiver' turbine positioned in the centre; polar values in the blue visualise the proportion of samples for a given wind direction that were classified as flow with a clear wake deficit. These proportions are presented on the plot as radial values between 0 (all samples in a

simulation identified as 'no detectable impingement') and 1 (all samples in a simulation identified as 'full impingement', 'partial impingement right' or 'partial impingement left').

Examining both Fig. 14 and Fig. 15, most plots depict the wakes generated by the upstream turbines as correctly identified for the corresponding wind directions at the 'receiver' turbine. There is a distinct difference in the proportion of samples identified as a wake depending on the distance between the 'emitter' and 'receiver'. For the turbines further away, the proportion

of detected impingement is lower due to the wake losing its distinctive shape when developing over a longer distance and interacting with the turbulent ABL for a longer time. This relation is apparent not only in the number of 'impingement'

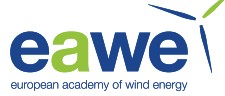


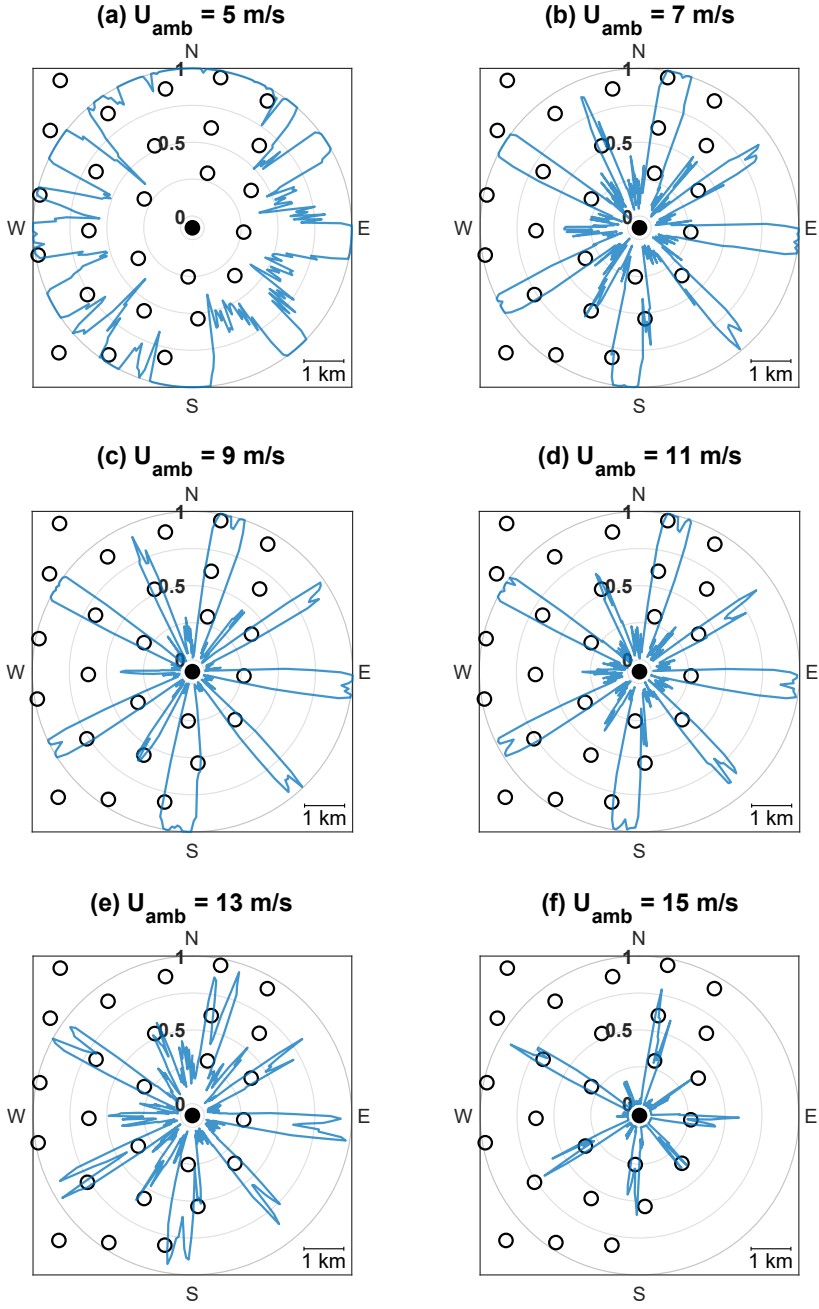

**Figure 14.** The wake detection model performance tested under a full wind direction range for six $U_{amb}$ values between 5 and 15 m/s. The radial plot in the blue indicates the proportion of samples classified as wake impingement for a given wind direction. Wind turbine locations indicated with black circles.



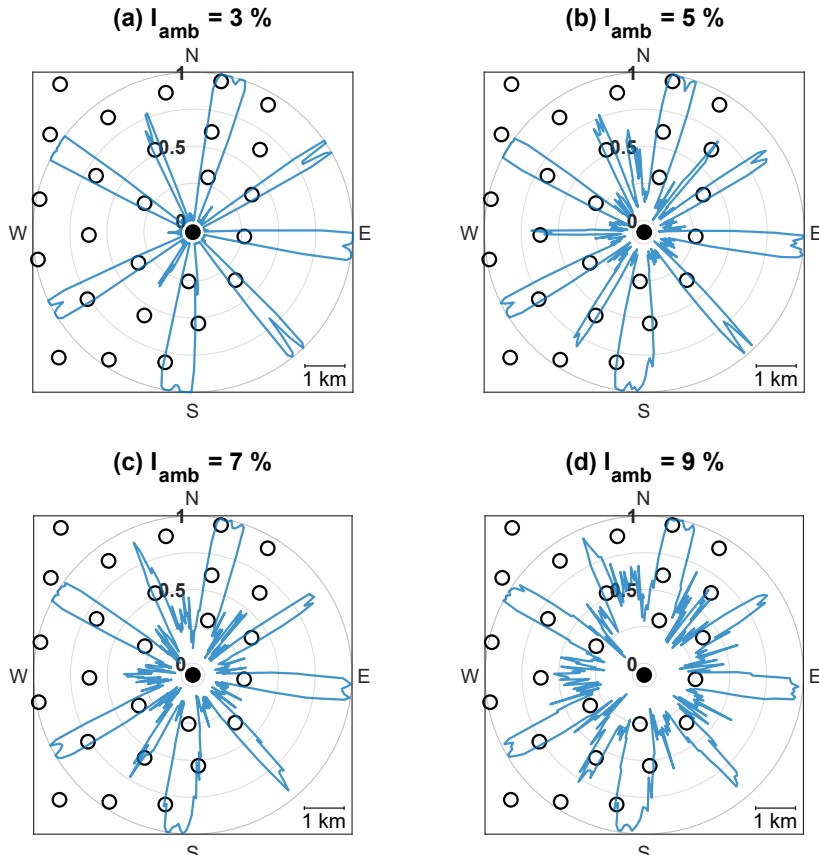

**Figure 15.** The wake detection model performance tested under a full wind direction range for four $I_{amb}$ values between 3% and 9%. The radial plot in the blue indicates the proportion of samples classified as wake impingement for a given wind direction. Wind turbine locations marked with black circles.

samples per simulation (displayed as the length of the blue marker 'tail') but it is also pronounced in the number of wind directions that the wake was detected for (the width of the blue marker 'tail').

The sensitivity analysis shown in Fig. 14 indicates that the best performance can be observed for $U_{amb}$ values of 7, 9 and 11

m/s. The detected impingement values have a minor noise (up to 0.3 impingement detected for all wind directions), nevertheless the wakes from all of the nearby turbines are appropriately identified with distinctive values close to 1 occurring for several wind directions each. For $U_{amb} = 5$ m/s there is a large amount of noise; in fact, for those of the S and E wind directions where there are no turbines upstream, the impingement was detected for approximately half of samples. This phenomenon appears to be caused by the wind sensing deficits observed in very low wind speeds made during the RMSE calculations. The majority of

other wind directions report impingement proportions of approximately 1, dropping to lower values for only few angles. For $U_{amb} = 13$ m/s and $U_{amb} = 15$ m/s, there are small dips visible in the centre of impingement proportion markers, causing them





to have a slight 'V' shape. A manual inspection of the results indicated this is can occur if a large and centered wake covers nearly all of the impinged rotor, presenting similarly to an unimpinged case. Overall, with the exception of $U_{amb}$ = 5 m/s, there is an inversely relationship between the mean ambient wind speed and responsiveness of the wake detection model.

Figure 15 shows how the wake detection performance is impacted by varying turbulence intensity $I_{amb}$ in the atmospheric flow. The ambient wind speed $U_{amb}$ was here kept at a constant value of 10 m/s. Firstly, it is apparent that increasing the turbulence intensity results in increased noise; this is manifested by having an increasing minimum amount of impingement detected for all of the wind directions for (b), (c) and (d) cases, with values of approx. 0.1, 0.15 and 0.3 respectively. On the other hand, the $I_{amb}$ = 3% case doesn't show significant impingement for turbines located farther away, such as the one directly

west of the 'receiver'. This contrasts with the behavior observed in the $I_{amb}$ = 5% case, where more pronounced impingement is reported for turbines at similar distances. For the higher $I_{amb}$ at (c) and (d), the detected wake impingement for turbines at respective wind directions is lower, which could be due to stronger mixing in the atmospheric flow.

The (d) subplot in Fig. 15 shows the case of ambient turbulence intensity equal to 9%. This is a unique case, as the training data (for both wind sensing and wake detection estimator models) only contained samples with $I_{amb}$ = 3, 5 or 7%. For this

reason, the results from this simulation set are a way of presenting how applicable the framework is to working with wind fields that are significantly different from the training data. It can be seen that while the largest impingement proportion is reported at the correct positions, indicating the wakes from nearby turbines correctly, there is now a significant amount of noise for all wind directions.

### 4.3    Wake detection and characterisation: visual results

Figures 16 and 17 present the performance of the framework in a more detailed manner than discussed in the Sect. 4.2. In total, results for six wind fields are showcased, with two subplots each. Firstly, the top plots show direct wake detection outputs from the CNN's classification. For each sample, a value between 0 and 1 is assigned to each of the four classes, representing the probability that a given sample depicts a corresponding wake impingement case. These four probabilities have a total sum of 1, due to which when the sample is a near-perfect representation of a class, the corresponding confidence score is near 1, while

the three other values are near 0. Secondly, the bottom plots report the outputs of wake characterisation, presenting the fitted and filtered lateral wake properties. These were plotted with the simulation wind field shown in the background, providing an easy way to assess the accuracy of the framework.

Cases (a) - (c) in Fig. 16 present varied examples of wind fields where the framework was operating with wake impinged flow and it successfully characterised the wake properties. The figure shows how the proportion of successfully detected and

characterised snapshots decreases with rising $U_{amb}$. The time series of fitted wake centre and border is continuous for (a), and progressively becomes disconnected when moving to (b) and (c). For all three cases it can be seen how the CNN's output aligns with the respective wind slice samples; for example, when wake meanders from the centre to the upper part of the Y axis, the highest probability transitions from 'Full impingement' to 'Partial impingement left'. Moreover, wake characterisation is not performed for the short periods of 'No detectable impingement' class. As seen in (a), due to low flow turbulence there is almost

no wake meandering, resulting in near-constant detection and characterisation outputs. Looking now at (b), it can be seen how

**Figure 16.** Detailed wake detection (top subplots) and characterisation (bottom subplots) results. All three wind fields with clear wake deficit present. (a) $U_{amb} = 7$ m/s, $I_{amb} = 5\%$; (b) $U_{amb} = 10$ m/s, $I_{amb} = 5\%$; (c) $U_{amb} = 13$ m/s, $I_{amb} = 5\%$.

the meandering motion is visibly increased, while the framework continues to accurately capture the wake development. The structured deficit dissipates toward the end of the simulation (after t = 450 s); this leads to a brief pause in wake characterisation, followed by a resumption with a significantly broader characterised wake. Moving on to (c), it can be observed how the ambient turbulence causes the structured deficit to break up, losing the continuity of characterised properties. There are some easy ways to remedy that issue (e.g. Kalman filtering), which provides a basis for future work.



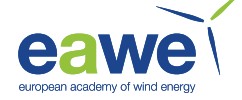


**Figure 17.** Detailed wake detection (top subplots) and characterisation (bottom subplots) results. (a) Highly dissipated wake deficit, $U_{amb} =$ 5 m/s, $I_{amb} = 9\%$; (b) Intermittent partial wake deficit, $U_{amb} = 15$ m/s, $I_{amb} = 5\%$; (c) No wake deficit, $U_{amb} = 5$ m/s, $I_{amb} = 5\%$.

Figure 17 presents examples of wind fields which for different reasons proved to be more challenging for the wake detection model. Case (a) at Fig. 17 presents an example of a wind field where due to high turbulence, the wake deficit is not sufficiently structured and the characterised properties are not continuous. Another consequence of the high turbulence is the more noticeable lag resulting from the moving average filtering; it can be seen that the characterised wake is shifted with respect to the corresponding wind field. Case (b) at Fig. 17 presents an example of a wind field where high atmospheric turbulence causes the wake to intermittently meander into the captured Y range. It can be seen that the CNN classified the majority of





samples as 'No detectable impingement' and the wake characterisation was not initiated. At time stamps of approx. 50, 350, 400, 450 and 500 seconds a wake deficit was detected, resulting in a short-term fitting of wake properties. With such a low quantity of identified impingement samples, the characterised wake properties end up being disconnected. Case (c) at Fig. 17 presents a wind field with no turbines upstream from the 'receiver'. It is apparent that due to low turbulence and wind speed, the framework mistakenly classified ambient fluctuations as wake impingement. The wind field presented here is the same as the one shown at Fig. 13 (a); a careful analysis allows to determine that the minor deficit noticeable at the 'Estimation' subplot matches the characterised wake centre and border from the Fig. 17.

## 5    Discussion

### 5.1    Current framework limitations

Beyond the methodological limitations outlined previously, it should also be highlighted that the application of a moving average filtering within the methodology results in predictions lagging behind the true wind field. This lag ranges from approximately 12 seconds for $U_{amb}$ = 15 m/s, to approximately 58 seconds for $U_{amb}$ = 5 m/s. These values are a consequence of matching the filtering window width to the wind field's typical meandering time scales (see Eq. (6) and (7)). In the context of wind farm flow control, these levels of lag aren't necessarily problematic, and the lag may even be removable by undertaking a short-term forecast across the gap. Kalman filtering would likely be a relatively straighforward way to achieve this. Implementing other approaches such as Long short-term memory (LSTM) networks could potentially allow for the forecast to be extended to predict wake locations and meandering behaviour a few minutes ahead. Further research needs to be conducted to investigate these leads.

The wake detection results captured at Fig. 14 and Fig. 15 indicate that the higher ambient turbulence and wind speed introduce considerable noise and reduce the size of 'wake markers', meaning that portions of samples were misclassified. Similarly, the wake characterisations obtained during the testing contain several gaps for higher turbulence and wind speed values. However, these issues should be considered less critical after the consideration of the specific application of this work. Firstly, the wind farm flow control brings largest benefits for the below rated operation, where due to higher energy extraction and lower wind, the wakes are more pronounced (Scott et al., 2024). These conditions align with the scenarios where the framework demonstrates optimal performance. Moreover, the wind speeds above 13 m/s (where the framework performance deteriorates) are generally less frequent than lower-wind conditions (Shu and Jesson, 2021), further mitigating the impact of these gaps.

Note also that the range of ambient turbulence intensities used for framework testing fell between 3% and 9%, and the atmospheric conditions for both training and testing were considered as neutral. Large offshore wind farms could experience higher levels of turbulence deep within the turbine grid due to multiple wake interactions (Shaw et al., 2022). These simplifications should be acknowledged and other turbulence levels, along with incorporating both stable and unstable ABL conditions, analysed in future work. Similarly, turbine yaw offsets were also not considered in the this work. The current analysis implemented



a medium-fidelity wake model, and so further work should also extend this framework to more realistic wake structures, for example those obtained through LES.

## 5.2 Framework applicability

Overall, the framework has proved to work well for the majority of tested ambient wind speeds and turbulence intensities. Although there were minor discrepancies for specific wind conditions, the developed wind sensing model presented generally good performance with low error between true and estimated values. The overall average RMSE for wind sensing across all cases was 0.35 m/s, or 5.23% when normalised by respective $U_{amb}$ values. This in turn allowed for good wake impingement analysis based on image recognition. With regards to wake detection, the framework's responsiveness to impingement proved to be dependent on $U_{amb}$ and $I_{amb}$ of the wind field; consequently, the following wake characterisation was sometimes compromised due to wrong classification of some of the samples. The selected bivariate Gaussian wake profile can easily adapt to cases where the wake is 'stretched' due to horizontal sheer or veer. The present methodology and analysis treats each 2D wind snapshot entirely independently, it therefore seems clear that improved performance (and removal of some of the problematic cases which have been identified) could be readily achieved by extending the methodology to undertake a post-processing analysis which account for time variations in results. This could, for example, allow for the imputation of gaps that occur in characterised time series for more turbulent wakes (see Figure 16); moreover, by considering classifications of time-adjacent snapshots, it could help in correct detection of those fully impinged cases which have been misidentified as unimpinged. Again, Kalman filtering would seem to be a technique well-suited to these possible methodological extensions.

The presented results also highlight that wake detection performance is highly dependent on the training data, indicating that improved performance across a broader range of conditions can be achieved by enlarging the conditions comprising the training dataset. Overall, the presented performance shows that the developed method for generalised wake impingement estimation is a promising solution to this challenging problem.

## 6 Conclusions

The proposed method for generalised wake impingement detection and characterisation shows good performance for a wide range of tested wind conditions. The framework's effectiveness is sensitive to ambient wind speed and turbulence intensity levels, with optimal performance observed for wind speeds between 7 and 13 m/s and turbulence intensities ranging from 3% to 9%. This work provides a baseline concept for generalised wake impingement estimation, which, with further improvements, could greatly contribute to better-informed wind farm flow control. Next steps for the futher development of this framework include: validation of the current setup in progressively more realistic environments (including yaw misalignment, higher fidelity wind fields and wakes, or utilisation of field data); an extension of the training dataset (towards accounting more flow phenomena from more turbines at different distance); methodological extension to capture time-variance between individual 2D snapshots and opportunities for imputation and forecasting of wake properties. Kalman filtering was identified as a likely route to extending the framework in this manner.





## Appendix A:  Convolutional Neural Network

The Convolutional Neural Networks (CNNs) are a specialized type of deep learning method, optimised for working with grid-like data; most commonly 2-D images, but they can also be used for 1-D or multi-dimensional data. Their design is perfectly suited for learning spatial features from input data, making them especially effective for image recognition and classification tasks. Due to several key achievements in recent years, they have revolutionised areas such as face recognition, hand writing analysis or autonomous vehicles (Li et al., 2022). In the context of wind energy, some examples of CNN implementation include wind power prediction (Zhu et al., 2017; Harbola and Coors, 2019), early fault detection and classification (Rahimilarki et al., 2022) and others. The input layer of the CNN is the size of the grid-like data it is processing - in a simple example of a grey-scale image, it would be its width multiplied by height. If the task at hand is to classify the image, for example decide on what handwritten digit does the image show, then the output layer would consist of 10 nodes (one for each digit), each one holding the confidence value between 0 and 1. The actual performance happens in the hidden layers located between the input and output layers, and for most CNNs, that is a combination of following:

- *Convolutional Layer*: applies learnable filters (also called kernels) that slide across the input data, only working on a small patch of the data at a time. A dot product of kernel and data at each position creates a measure of how close a patch of data resembles a given spatial feature like an edge or an arch. Doing that for the entire input results in a feature map. Performing that operation through multiple iterations with many kernels, progressively optimizing their weights, allows to combine small features into larger ones and ultimately is the key to detecting patterns.

- *Activation Function*: normally following the convolution layers, various activation functions like ReLU (Rectified Linear Unit) introduce non-linearity, enabling the network to learn complex patterns. It is achieved by restricting the connections between the neurons when the weighted sum is too low.

- *Pooling Layers*: reduces the spatial size of feature maps, decreasing computational complexity and helping to prevent overfitting. Max pooling is the most commonly used method of doing so, selecting the maximum value from each patch of the feature map.

- *Fully Connected Layer*: after several cycles of convolution-activation-pooling, this layer connects every neuron in one layer to every neuron in the next, combining features learned by previous layers to produce final predictions.

While designing the neural network, one needs to decide on the number and type of hidden layers implemented in its architecture, as well as on the hyperparameters such as number and size of learnable filters. Moreover, as with all deep learning techniques, the performance of the CNN is heavily dependent on the quality of data used for its training.

## Appendix B:  A comparison of the univariate and bivariate 2D Gaussian fit

A particular, univariate case of the 2D Gaussian function occurs when there is no correlation ($\rho = 0$) between Y and Z and when the variations in both axis are assumed to be equal - in that case, $\sigma_y = \sigma_z = \sigma$ and thus the wake is considered to be a





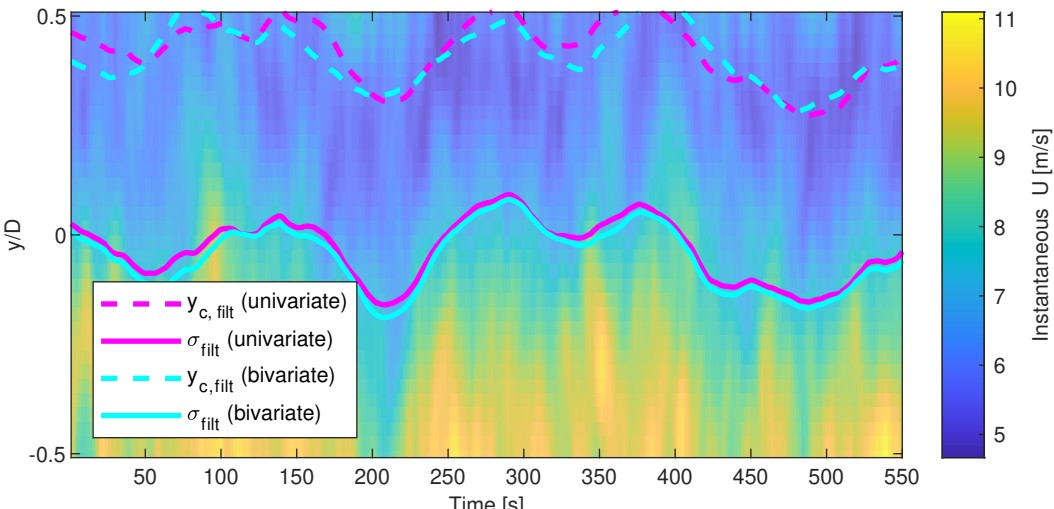

**Figure B1.** A comparison of filtered wake characterisation output time series for univariate and bivariate 2D Gaussian fitting applied. Can be imagined as a view from above - 'true' wind field shown in the background analogically to Fig. 12 and 13.

circle. The function is simplified as follows:

$$f_{G,univariate} = \frac{A}{2\pi\sigma} \exp\left[-\frac{1}{2}\left(\frac{(y_i - y_c)^2 + (z_i - z_c)^2}{\sigma^2}\right)\right] \tag{B1}$$

As a result of using a univariate function, the number of parameters to optimize is reduced from six to four: $(A, \sigma, y_c, z_c)$. Both univariate and bivariate fits were applied to several varied wind fields in search for differences in the wake characterisation performance in this specific application. Generally, it was found out that due to the moving average filtering applied to the fitted wake centre and lateral width time series, the ultimate performance of wake characterisation did not significantly change when switching from one variant of the 2D Gaussian function to another. This effect was captured on an example wind field presented in Fig. B1. It can be noticed how both estimated wake properties follow approximately the same path. Figure B2

shows a selected YZ slice from the same wind field viewed from the front, where the differences in fitted wake shapes can be easily noticed. The fitted univariate function encapsulates the regular, circular deficit shape, while the bivariate algorithm fits a function that has one of the spreads significantly larger, hence resulting in an ellipse. For that reason, the univariate function is unable to represent cases where the wake is skewed due to turbulence, shear or veer. With that in mind, to allow for a better fit and more accurate wake characterisation results, the bivariate function was implemented in the project.



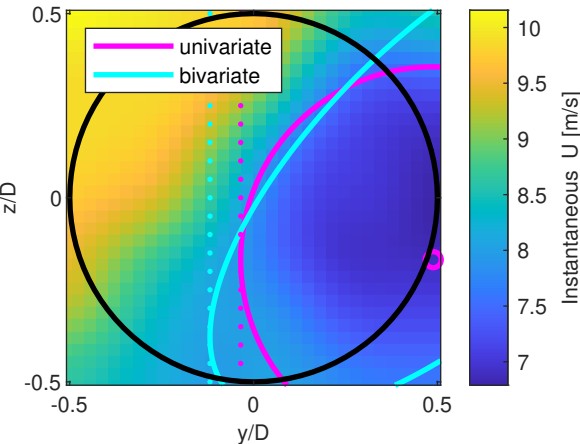

**Figure B2.** A comparison of wake characterisation performance for a single YZ slice with univariate and bivariate 2D Gaussian fitting applied. Rotor outline marked with black line; fitted circle/oval marked with magenta/cyan solid line; maximum y point (signifying the lateral width parameter) marked with a vertical magenta/cyan line; fitted wake centre for the univariate variant at $(y_c, z_c)$ marked with magenta circle; fitted wake centre for the bivariate variant not visible due to being outside of the snapshot. 'True' wind field YZ slice shown in the background.

*Author contributions.* Conceptualisation: Fojcik, Hart, Hedevang. Writing (original draft): Fojcik. Writing (review and editing): Fojcik, Hart, Hedevang. Analysis: Fojcik. Visualisation: Fojcik. Supervision: Hart, Hedevang.

*Competing interests.* The authors declare that they have no conflict of interest.

*Acknowledgements.* This research has been funded by Siemens Gamesa Renewable Energy and EPSRC Industrial CASE program (grant no. EP/W522260/1, voucher no. 210039).





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
