# Peer review of "Wind turbine wake detection and characterisation utilising blade loads and SCADA data: a generalised approach"

_Wind Energy Science, 2025_

## Referee Comment (RC1)

**Review: *Wind turbine wake detection and characterisation utilising blade loads and SCADA data: a generalised approach***

**General Comments**

This article presents a model for the detection and characterisation of wakes via estimating wind fields from blade load data. The model is well-explained, and the paper reads very clearly, with informative visualisations of the results. The approach developed here would be of interest to readers of this journal. There are a few areas that could be expanded on, including a deeper look into the accuracy of the wake impingement classification and wake characterisation. Additionally, a detailed flow-chart of the full process would guide others looking to reproduce these results.

**Specific Comments**

1. Metrics / Accuracy:
   a. Is there a metric by which it is decided whether a wake is impinging on a rotor, e.g. a velocity deficit threshold? In line 39, "significant" impingement in terms of both magnitude and time is mentioned, but there is no further detail on the training data labelling for wake detection and classification. Could the details of how flow was classified as containing a wake be included in e.g. Section 3.4?
   b. For the reported wake detection accuracy of 91% in Section 3.4, did this vary by "class" of impingement, e.g. was the model more or less accurate at predicting partial impingements? This may be relevant for future work using this model in wake steering controllers.
   c. It would be informative to include accuracy metrics of related wind flow estimators or wake classifiers, to provide context for the model(s) developed in this paper.
2. Results:
   a. When the trained model was tested on a new receiver turbine in Section 4, were metrics for the accuracy of the wake detection or classification models calculated? In particular for the wake impingement predictions under 9% turbulence intensity, were the results in e.g. Figure 15(d) confirmed to be sensible given the increase in turbulence compared to training data?
   b. Is there an explanation for the "fake" wakes seen in Figure 17(c) / 13(a), or a proposed method to alleviate this? These simulated areas were mentioned as potentially resulting in mis-classification, is there any way to quantify how often this might occur?
   c. Did the superposition of wakes or the position of the turbine deep within the farm have any effect on model's accuracy in Section 4?
3. Flow Chart:  It would be very useful to have a more detailed flow chart (i.e. more in-depth version of Figure 3) that includes the steps take for e.g. pre-processing to extract wind field from turbine blade loads, fitting of DCT factors, constructing wind fields, sampling frequency and fitting wake parameters.
4. Pre-Processing Diagram: A diagram of the turbine loads and how they are transformed would be informative in Section 3.3.
5. CNN Model: More detail on the CNN architecture is needed in Section 3.4.

6.  Conclusions: The conclusions are very brief, they should be expanded to include a summary of the accuracy of the models developed, as well as a short description of current limitations before future work.

**Technical Corrections**

1.  General: Please ensure all acronyms are defined with capitalisation at the first use, and used consistently thereafter.
2.  General: Please be consistent with using either double or single quote marks.
3.  Line 25: Please clarify "the aforementioned task".
4.  Line 27: "altered" does not give enough information about the features of waked flow that lead to higher loads, suggest re-wording to e.g. "experience a **more turbulent** wind field".
5.  Line 34: "yaw control" usually refers to control of a single turbine to follow the inflow, the standard term for farm-wide yaw optimisation is "wake steering"; suggest using this term instead.
6.  Line 50: "**to date**"
7.  Line 52: Typo: "turbine**'s** wake"
8.  Line 65: Suggested re-word: "that **the** focus of the current work is **to develop** a solution"
9.  Line 87: Please include a reference for the microscale length scale.
10. Figure 1: I think it should be $A_1 = A_0/(1-a)$?
11. Line 109: "as **a** *wake*"
12. Line 110: Please include a reference for the "2-4 rotor diameters" statement.
13. Line 119: Please include a reference for the Gaussian wake model relations.
14. Line 125: I think "**differs**" is meant rather than "defers"?
15. Line 135: The explanation around atmospheric shear and the location of maximum turbulence needs more detail.
16. Line 145: For the infinite wind farm case, the power extraction from the turbines is balanced by entrainment of kinetic energy from the flow above; the explanation given here seems to reference increasing vertical height in the ABL?
17. Lines 153 & 154: Unclear wording, is the data from the first 10 rows and first 8 rows of turbines per farm? And is the power loss between 45% and 70%?
18. Line 181: Suggested re-word: "distinguishes the **various impacts** of turbulence"
19. Line 187: Suggested re-word: "The **widely-discussed method** introduced by..."
20. Line 196: Suggested re-word: "in incoming **flow**"
21. Line 217: The phrase "accurate approximate estimation" does not make sense.
22. Line 225: "wind farm **flow** control" for consistency.
23. Line 264: This sentence is quite convoluted, please re-word.
24. Line 276: What kind of evaluation metrics were used to determine the model had reached sufficient accuracy?
25. Line 298: Please specify whether "left" and "right" are as seen looking at the front or the back of the turbine.
26. Line 314: A brief description of conditional dependence would be useful here.
27. Line 315: Typo: "**the** following"
28. Line 348: Typo: "the the"
29. Line 354: "with **a** few"

30. Line 358: Could a brief list of all the inputs be given, either in the text or as a table, for clarity on what the 96 variables are?
31. Line 361: Suggested re-word: "are  processed"
32. Line 382: More description of the "simple models" is needed.
33. Lines 396 & 411: It would be easier to read the proportions than the actual numbers of simulations, e.g. 10% instead of 1,120 on line 411.
34. Line 414: Suggested re-word: "case **was** approximately 91%"
35. Line 430: Definition of the "rotation angle" needed.
36. Line 433: "D" has already been used as dimension e.g. "2D".
37. Equation 6 (Line 446): Is the integral missing $dt$?
38. Section 3.6: This would make more sense as the first part of Section 4.
39. Line 460: Suggested added wording: "centrally located **within the wind farm**"
40. Line 480: "**have** a low mean RMSE"
41. Line 503: Suggested re-word: "simulations **showed** that"
42. Line 505: The term "amount of turbulence" is ambiguous, since the turbulence intensity has not changed but the wind speed has increased. Please re-word for a clearer explanation.
43. Line 527: Suggest using "**South and East**" rather than "**S and E**".
44. Line 534: Typo: "**inverse**"
45. Line 598: Suggested re-word: "after  consideration".
46. Line 599: Suggested re-word: "Firstly,  wind farm flow control brings **the largest benefits for below** rated operation"
47. Line 620: Suggested re-word: "(and **solution to** some of"
48. Line 643: "**2D**" and "**1D**" without hyphen for consistency.

---

## Referee Comment (RC2)

**Paper Title:**

*Wind turbine wake detection and characterisation utilising blade loads and SCADA data: a generalised approach*

**General summary:**

The manuscript presents a 3-stage methodology for turbine-based wind sensing, wake detection and wake characterization. Core of the methodology is a deliberate combination dimensionality reduction and machine learning methods, linking turbine data and wind field information. Both training and testing use aeroelastic simulations coupled with the dynamic wake meandering model and Mann turbulence. The wind sensing shows convincing results, both qualitative and quantitative. Wake detection and characterisation are mainly assessed qualitatively and show mostly good performance. The shortcomings of the methodology and possible improvements are discussed.

**General comments:**

1. The abstract is good and expressive!
2. Regarding the background chapter: The paper should be concise and focus on its main topic. A generic literature review of >6 pages is not appropriate in this context, especially since these topics are not picked up in the discussion section. The target audience can be expected to have a wind energy background. The interested reader will not choose this paper to learn about ABL, momentum theory or wake physics.
   Bottom line: It's suggested to remove section 2 "Background" and include the literature review with relevance to the paper topic (mainly contained in subsections 2.4 - 2.6) in the introduction section. (Accordingly, no specific comments were done for section 2 at this point).
3. The manuscript mixes past and present tense (e.g. in sections 3.2, 3.4, 3.5.2, 5.2). Please formulate in present tense where possible.
4. The language could be more concise. There are many instances of "As explained earlier, …", "First of all, …", "With that in mind, …", etc.
5. At many instances in the paper, simulation parameters and numbers of runs are mentioned (training/sensing tests/performance tests). Gathering all that information in one concise table would be helpful.
6. The manuscript has individual "Results" and "Discussion" sections, which is good. Yet, the results section already contains aspects that would belong in the discussion section (e.g. Line 488-496, 520-521, 541-542, 569-570). On the other hand, the discussion section is very brief and further lacks a comparison to existing methods. Please revise and make sure to have a clear distinction, possibly ending up with a shorter but more concise results section and a more in-depth discussion section.
7. In section 4.2 and 4.3 the wake detection is mainly assessed qualitatively and visually. The detection ratio (in Fig. 14&15) is the proportion of detected wakes out of all sample slices, but not with respect to a reference. Using the DWM model, the wake positions of the simulations should be available. It is suggested to use this information and show a quantitative performance metric of wake detection. Furthermore, it's suggested to use the RMSE of the estimated wake position as a metric of the characterization. Additionally, this information could help to unravel the unexpected behaviour of the detection at 5 m/s in Fig. 14.

**Specific Comments:**

*Abstract*

1. Line 9: "virtual wind farm" – Please state the test environment here. It should be clear from the beginning how the method is tested. Especially since the title does not tell whether it's in field/simulations/wind tunnel.

2. Line 15: Partial wake conditions are not necessarily harder to track. The high load-imbalance along the rotor can even make it easier to estimate in comparison to a full wake. Your results, e.g. the findings of Fig. 11, do not seem to support this statement. It's suggested to leave out that sentence.

*Introduction*

3. Line 19: "at the dawn of 2023" – please consider replace by "by end of 2023"

4. Line 35/36: Cyclic pitch control is just one type of "dynamic induction control", which should be mentioned here as the general discipline. It splits up into Pulse and Helix. Frederik et al. focus on Helix. Please adjust and add a more general source.

5. Wind farm flow control techniques are mentioned in the introduction, but the connection to the tracking task within the framework of closed-loop control is lacking. The outlook and final role of the wake tracking and characterization should be mentioned.

6. Line 49-53: This is not entirely true. The approach of (Bottasso et al., 2018) is used for impingement detection and EKF-based approach in (Onnen et al., 2022) further links wake-presence to the observability. Yet, the 3-stage approach of this manuscript is a novelty and the justification for initial unbiased wind field reconstruction, as mentioned in line 57, is there. Please elaborate in the paragraph and differentiate between the approaches.

7. Line 55-56: "For this reason, the vast majority of methods developed so far are not yet applicable to real world operations." This is a too strong statement, considering that there are field validations for these other methods, see (Schreiber et al., 2020), (Lio et al., 2021). As said in the previous comment: The research gap exists, but it is not accurately described in this introduction.

8. Line 65-67: "It is highlighted that focus of the current work is that of developing a solution which is able to confidently assert when a turbine is impinged by a wake from a nearby turbine, as this information is critical to farm level wake steering control." – Please rephrase this sentence or make it two sentences.

9. Line 64-65: "A full end-to-end methodology is presented, which aims to provide both a demonstrator and performance benchmark for generalised wake detection and characterisation methods of this type." – This would be a good place to mention the test environment (aeroelastic simulations, turbine type, DWM model, …).

*Methodology*

10. Line 295: why wind farm simulations, when only two turbines are used? Relating to intro: "generalized approach"

11. Line 290-295: Please state the turbine type, diameter and the spacing between "emitting" and "receiving" turbine.

12. It is unclear, whether the simulation environment includes just these two turbines or the whole wind farm.

13. Line 309: "Figure 6 illustrates the process for training the wind sensing model and demonstrates its post-training performance in producing wind field estimations." The application scheme of the model is shown here, but not post-training performance. Please adjust the sentence.

14. Eq(1) & Eq(2) (and possibly others): don't use italic font for sine and cosine and subscript text (except variables).

15. Line 344ff: This is the first time that higher harmonics are mentioned. The Coleman transform was only described for the 0P / 1P harmonics. Also: Fig. 7 names an "original load", which suggests a (non-transformed) blade load. Meanwhile, it's said in line 344 that the rotor loads are decomposed into their frequency components. Is it correct, that you e.g. calculate the 3P share of a yaw moment? Or do you calculate the 3P of a blade load? A flow chart of the pre-processing steps would help.

16. Line 351ff: Please elaborate on the temporal dependency and time lag. Which temporal scales of the turbine dynamics are you addressing here?

17. Subsection 3.3.3: The DTC is a nice choice and the dimensionality reduction shown in Fig.8 looks appropriate. My only point regarding the wind field parametrization is: The here shown YZ-wind field slices are rectangular, while the rotor swept area is circular. The corners of the wind field thus include non-observable features, but could still influence the lower-order share of the DTC outputs. Were the plain rectangular slices used for the training? Or was any weighting or masking applied? Please comment on whether you expect an impact here.

18. Section 3.3.4: Please add some more details and a literature source to the used regression approach.

19. Line 392ff: Is the distinction of the four classes based on the constellation of the simulation run or based on the instantaneous wake position (which could differ due to the employed DWM model). Also: Please define the overlap margins, from which you categorize full/partial/no wake impingement.

20. Section 3.5.1: The fitting function does not fully reflect the fit that was probably implemented. To fit a wind field with wake deficit, it should be U = u_ambient – f_G, here considering that parameter *A* is negative.

21. Section 3.5.2: How does the low-pass filter deal with falsely identified no-impingement instances?

22. Line 456: Please rephrase this sentence.

23. Fig. 11: Please add more details to the figure caption.

24. Section 4.1: a diff-plot between estimated and simulated wind field would help to analyse, whether systematic or just random differences exist (e.g. the central deficit mentioned in line 500)

25. Line 501-503: "A slight disagreement would not be normally problematic; however, in this case where the flow has little overall turbulence, a subtle deficit like this can't 'hide' among other eddies, which could potentially result in classifying the mentioned samples as being wake impinged." This sounds rather complicated and nested. Please rephrase the sentence.

26. Fig. 14 shows that simulations for all wind directions are on hand. Please report this more explicitly in section 3.6.

27. Fig. 14 @ 5 m/s: why is detection ratio different here? In partial load range, the non-dimensionalized wake should be similar, thus limited impact on the sensing is expected. Also: It would be good to know the turbine's cut-in wind speed. At 5 m/s undisturbed ambient wind speed, a wake-exposed turbine might experience a rotor-effective wind speed below cut-in.

28. Fig. 17 b&c) especially here it would be helpful to see the wake position reference from the simulation environment (see general comment 8).

*Discussion*

29. Line 591ff: ". Implementing other approaches such as Long short-term memory (LSTM) networks could potentially allow for the forecast to be extended to predict wake locations and meandering behaviour a few minutes ahead. Further research needs to be conducted to investigate these leads." Please provide a source here and a stronger supporting argument for the claim that the (stochastic) meandering wake location can be predicted by the receiving turbine. Otherwise, please consider softening this statement.

*References*

30. Line 700: incomplete reference (journal, DOI)

**References**

Bottasso, C. L., Cacciola, S., & Schreiber, J. (2018). Local wind speed estimation, with application to wake impingement detection. *Renewable Energy*, *116*, 155–168. https://doi.org/10.1016/j.renene.2017.09.044

Lio, W. H., Larsen, G. C., & Thorsen, G. R. (2021). Dynamic wake tracking using a cost-effective LiDAR and Kalman filtering: Design, simulation and full-scale validation. *Renewable Energy*, *172*, 1073–1086. https://doi.org/10.1016/j.renene.2021.03.081

Onnen, D., Larsen, G. C., Lio, W. H., Liew, J. Y., Kühn, M., & Petrović, V. (2022). Dynamic wake tracking based on wind turbine rotor loads and Kalman filtering. *Journal of Physics: Conference Series*, *2265*(2), 022024. https://doi.org/10.1088/1742-6596/2265/2/022024

Petrovic, V., Jelavic, M., & Baotic, M. (2014). Reduction of wind turbine structural loads caused by rotor asymmetries. *2014 European Control Conference, ECC 2014*, 1951–1956. https://doi.org/10.1109/ECC.2014.6862484

Schreiber, J., Bottasso, C. L., & Bertelè, M. (2020). Field testing of a local wind inflow estimator and wake detector. *Wind Energy Science*, *5*(3), 867–884. https://doi.org/10.5194/wes-5-867-2020

---

## Author Comment (AC1)

Thank you for taking the time to review our manuscript and for your valuable comments, which have helped us enhance the quality of the paper. Below, we include your comments in **black**, followed by our responses in **blue**.

**Paper Title:**

*Wind turbine wake detection and characterisation utilising blade loads and SCADA data: a generalised approach*

**General summary:**

The manuscript presents a 3-stage methodology for turbine-based wind sensing, wake detection and wake characterization. Core of the methodology is a deliberate combination dimensionality reduction and machine learning methods, linking turbine data and wind field information. Both training and testing use aeroelastic simulations coupled with the dynamic wake meandering model and Mann turbulence. The wind sensing shows convincing results, both qualitative and quantitative. Wake detection and characterisation are mainly assessed qualitatively and show mostly good performance. The shortcomings of the methodology and possible improvements are discussed.

**General comments:**

1. The abstract is good and expressive!

Thank you for your feedback. We are happy to hear you liked it.

2. Regarding the background chapter: The paper should be concise and focus on its main topic. A generic literature review of >6 pages is not appropriate in this context, especially since these topics are not picked up in the discussion section. The target audience can be expected to have a wind energy background. The interested reader will not choose this paper to learn about ABL, momentum theory or wake physics. Bottom line: It's suggested to remove section 2 "Background" and include the literature review with relevance to the paper topic (mainly contained in subsections 2.4 - 2.6) in the introduction section. (Accordingly, no specific comments were done for section 2 at this point).

We agree that the literature review could be more concise. We have removed the Sections 2.1 – 2.3 (original manuscript numeration) entirely, your argument that these concepts are not explicitly used in the rest of the paper is valid. Section describing the numerical wake models (2.4 in the original manuscript) was converted into a short description of the which can be now found in lines 132-135. As per suggestion, Sections

2.5 and 2.6 (original manuscript numeration) are condensed and integrated into the introduction.

3. The manuscript mixes past and present tense (e.g. in sections 3.2, 3.4, 3.5.2, 5.2). Please formulate in present tense where possible.

Agreed, the tense should be consistent across the paper. We have modified for the revised manuscript, all paper is now in present tense.

4. The language could be more concise. There are many instances of "As explained earlier, …", "First of all, …", "With that in mind, …", etc.

We believe that in some places the linking words like the ones you mentioned are helpful, as they allow the reader to connect the points in the narrative. However, considering the length of the paper, we agree that it could be more concise. For the revised manuscript, we have simplified and made it less wordy wherever possible.

5. At many instances in the paper, simulation parameters and numbers of runs are mentioned (training/sensing tests/performance tests). Gathering all that information in one concise table would be helpful.

In the original manuscript, for the sake of brevity, we aimed to summarise this information in the text wherever possible. However, we do agree that this information would be easier to digest in the table format. We believe that adding two tables – one showing the training simulation runs, and one showing testing simulation runs, could be better than one unified one. This is a justified by a different distribution of ambient U and I values in these two cases (e.g. 6 U values for testing, 100 U values for training), and additional column for training (describing the wake impingement case). We also think that the reader shouldn't have to jump several pages to see the summary of the information from specific part of the paper. In the revised manuscript, new tables are numbered 1 and 4, for training and testing, respectively.

6. The manuscript has individual "Results" and "Discussion" sections, which is good. Yet, the results section already contains aspects that would belong in the discussion section (e.g. Line 488-496, 520-521, 541-542, 569-570). On the other hand, the discussion section is very brief and further lacks a comparison to existing methods. Please revise and make sure to have a clear distinction, possibly ending up with a shorter but more concise results section and a more in-depth discussion section.

We acknowledge your feedback here and agree that the in-depth analysis should be rather kept in the Discussion section. We have revised the two sections: in the revised manuscript, Results is now shorter and containing only presentation of results, and Discussion has following subsections: Evaluation methods, Framework performance, Applicability and Current limitations. With regards to your comment about lacking

comparison to existing methods, we have answered this in the response to the other reviewer, which we cite below:

"Although we do agree that this additional metric would be informative, we believe that adding a discussion subsection that would satisfy your comment is outside of the paper scope. The main novelty of this work is the introduction of a modular approach to the wake estimation problem – a generalised framework being a combination of several models. The models that we implemented can be easily swapped for something more accurate, and we want to encourage the community to use their methodologies this way.

Furthermore, a direct comparison with other detection/ characterisation studies would be difficult due to fundamentally different assumptions. Other works that consider the wake detection aspect, such as (Onnen et al., 2022) or (Bottasso et al., 2018), do not test their solutions under a full range of wind conditions as we did. Existing simulation-based wake characterisation studies such as (Onnen et al., 2022) use different approaches for establishing the reference."

7. In section 4.2 and 4.3 the wake detection is mainly assessed qualitatively and visually. The detection ratio (in Fig. 14&15) is the proportion of detected wakes out of all sample slices, but not with respect to a reference. Using the DWM model, the wake positions of the simulations should be available. It is suggested to use this information and show a quantitative performance metric of wake detection. Furthermore, it's suggested to use the RMSE of the estimated wake position as a metric of the characterization. Additionally, this information could help to unravel the unexpected behaviour of the detection at 5 m/s in Fig. 14.

Thank you for your comment, we agree that quantitative analysis would make this study better. We have spent significant time looking into the version of the DWM model that we're using, attempting to apply your suggestion, here's our conclusions:

a) wake detection: when computing the wind field at the receiving turbine, multiple meandering wake deficit profiles are in general imposed on an otherwise clean "Mann box", including additional added turbulence determined by the shape of the deficit profiles. To combine multiple overlapping wake deficit profiles, some rule is applied pointwise, that is, independency at each relevant grid position in the wind field. The rule we used was "pick the maximum deficit at the point", but other rules are possible. As a result, the wake deficit profile that is applied to the Mann box does not necessarily have a simple shape with a well-defined location of maximal deficit. For majority of wind directions where the flow is coming from the inside of the wind farm, there are always some influencing turbines registered by the code – even if they are too far away to have a clear effect on the 'receiver' device. As a result of the above, a reference such as 'waked/not-waked for a given wind direction' is impossible to establish directly from the

simulations. This is partially dictated by the nature of our study - we have tested the detection performance for all wind directions, where the wake shedding turbines are at various upstream distances, hence the rate of wake breakdown/lateral displacement is also varied. Furthermore, to the best of author's knowledge, there isn't a widely-recognised definition of 'wake impingement' (e.g. by means of reduced power output) we could use here. As a work-around to this issue, we have provided a reference for the quantitative analysis in a 'synthetic' way. We've trained a new classifier analogically to the process described in Methodology, with the only difference being that the training dataset is derived from simulated wind fields, not the estimated ones. Without the bias from the wind field reconstruction, this classifier achieves approx. 99% accuracy under integrated testing, and its classifications are thus assumed to be precise enough to become 'ground truth' reference. Such a reference allows to consider the effects of varied wake dispersion under different ambient conditions, and arguably fits this analysis better than a general impingement definition based on inflow angle. All in all, the goal of our wake detector is to identify "clear wake impingement from a nearby device" as we define it in the article. The reference as seen in Fig. 12 (revised manuscript) are sensible, as the impingement ratio decreases with increasing wind speed – just what would happen in real life, as operating in the above rated region makes wake less pronounced. We consider the limitations and bias from this approach in the Discussion section.

b) wake characterisation: we have investigated how to best provide the reference for the estimated wake properties, and we've decided to establish it by fitting a 2D Gaussian function to YZ slices of the raw, simulated wind field. This approach is preferred over using the meandering wake centres applied internally in the DWM model, as turbulent fluctuations in the synthetic wind field — along with additional imposed turbulence – can cause the actual wake experienced by the turbine to deviate from the calculated position. Moreover, this method naturally accommodates the interaction of multiple, combined wakes. A similar approach for getting reference by fitting a Gaussian is used in other studies, like (Lejeune et al., 2022). We consider the limitations and bias from this approach in the Discussion section.

**Specific Comments:**

*Abstract*

1. Line 9: "virtual wind farm" – Please state the test environment here. It should be clear from the beginning how the method is tested. Especially since the title does not tell whether it's in field/simulations/wind tunnel.

Good point – we have revised the manuscript and added an appropriate sentence accordingly: "The framework is tested in a simulation environment incorporating the Mann turbulence model, DWM model for generating wakes and BHawC aeroelastic code."

2. Line 15: Partial wake conditions are not necessarily harder to track. The high load-imbalance along the rotor can even make it easier to estimate in comparison to a full wake. Your results, e.g. the findings of Fig. 11, do not seem to support this statement. It's suggested to leave out that sentence.

We agree, line 15 did contradict the results. We have cut out this statement from the sentence, replacing it with 'more turbulent conditions'.

*Introduction*

3. Line 19: "at the dawn of 2023" – please consider replace by "by end of 2023"

Sorted – we changed it to 'by the end of 2022', which is what the source used explicitly says.

4. Line 35/36: Cyclic pitch control is just one type of "dynamic induction control", which should be mentioned here as the general discipline. It splits up into Pulse and Helix. Frederik et al. focus on Helix. Please adjust and add a more general source.

The authors of a recent comprehensive review article on the subject (Meyers et al., 2022) describe the Pulse as dynamic induction control and Helix as dynamic individual pitch control. These two terms are also differentiated by the authors of the Helix method (Frederik et al., 2020). We believe it is valid to use different terms here: Pulse uses sinusoidally varying thrust coefficient (Munters and Meyers, 2018), which is why dynamic induction control fits perfectly; while Helix varies the tilt and yaw moment at the rotor, hence the induction factor is not directly modified. With that in mind, we have adapted the sentence towards: "(…) some examples include wake steering by introducing yaw offsets (Howland2020, Siemens2019), or dynamic individual pitch control (Frederik2020) and dynamic induction control (Munters2018) to induce enhanced mixing in the wake."

5. Wind farm flow control techniques are mentioned in the introduction, but the connection to the tracking task within the framework of closed-loop control is lacking. The outlook and final role of the wake tracking and characterization should be mentioned.

We do agree that the paragraph could be modified to better emphasise on the key role of wake 'sensing'. We have improved the narrative and added relevant source: "(…) To use the above mentioned approaches in a closed-loop control scheme, dynamic information on whether the impinging wake is being successfully redirected or dispersed is required (Raach2016). Moreover, before starting the control action, there

first needs to be a confirmation that a turbine is indeed wake-affected to facilitate an intervention to its normal operating cycle. For the sake of the discussion in this work, we will term these two flow control prerequisites wake detection and wake characterisation. By the former we refer to (...)"

6. Line 49-53: This is not entirely true. The approach of (Bottasso et al., 2018) is used for impingement detection and EKF-based approach in (Onnen et al., 2022) further links wake-presence to the observability. Yet, the 3-stage approach of this manuscript is a novelty and the justification for initial unbiased wind field reconstruction, as mentioned in line 57, is there. Please elaborate in the paragraph and differentiate between the approaches.

We do agree that our statement of research gap lacked accuracy in its original form. The sources you mention consider wake detection, which is why we shouldn't disregard their contribution to the field. Our main novelty is the generalised character of the framework, which combines relatively simple models to achieve unbiased wake estimation performance. These models could be easily replaced with other (more advanced and potentially better performing) solutions, if someone wanted to repeat our methodology. With that in mind, we have modified the entire introduction: a) to include the relevant literature review, which was initially in Background; b) to state the research gap more accurately.

7. Line 55-56: "For this reason, the vast majority of methods developed so far are not yet applicable to real world operations." This is a too strong statement, considering that there are field validations for these other methods, see (Schreiber et al., 2020), (Lio et al., 2021). As said in the previous comment: The research gap exists, but it is not accurately described in this introduction.

As mentioned above, we have significantly modified the Introduction section to address your feedback on the research gap statement.

8. Line 65-67: "It is highlighted that focus of the current work is that of developing a solution which is able to confidently assert when a turbine is impinged by a wake from a nearby turbine, as this information is critical to farm level wake steering control."– Please rephrase this sentence or make it two sentences.

The sentence was a bit convoluted, we have reworded it towards: "This work focuses on developing a novel solution that can confidently assert when a turbine is impinged by a wake from a nearby machine – this being a key factor for farm-wide wake steering control."

9. Line 64-65: "A full end-to-end methodology is presented, which aims to provide both a demonstrator and performance benchmark for generalised wake detection and

characterisation methods of this type." – This would be a good place to mention the test environment (aeroelastic simulations, turbine type, DWM model, …).

Agreed, added an appropriate sentence describing the overall training/testing setup: "The models are trained and tested for a full range of wind directions within a virtual offshore wind farm. The simulation environment incorporates Mann turbulence wind boxes, Dynamic Wake Meandering (DWM) model for generating wake interactions, and aeroelastic code to compute the turbine response."

*Methodology*

10. Line 295: why wind farm simulations, when only two turbines are used? Relating to intro: "generalized approach"

Indeed, at the current stage, only two turbines are used during training of the framework. Our focus was on isolating clear wake impingement conditions, so the capabilities of the image-recognition-based wake detection can be assessed. The fact that the models can be trained using only two devices (with data extracted only from the downstream turbine), should be considered an advantage when considering the field application. For example, if the 'true' wind fields for training would be acquired with LIDAR, the additional hardware would only need to be installed on one turbine. The entire wind farm layout with several turbines at once, and with wind coming from all directions, was used during the testing stage. We believe that this criterium justifies the use of wording 'generalised approach'.

11. Line 290-295: Please state the turbine type, diameter and the spacing between "emitting" and "receiving" turbine.

For confidentiality reasons, Siemens Gamesa prohibits us from publishing the details of the virtual wind farm/turbine we are using. As a result, we can't include the information on turbine type or diameter. We could however add the spacing between the turbines, expressed as a multiplication of diameter D. We included that in the revised manuscript in line 118.

12. It is unclear, whether the simulation environment includes just these two turbines or the whole wind farm.

The simulation environment includes the entire wind farm layout. Although we heavily rely on that aspect during testing, for training we purposefully choose turbines from first two rows and select specific wind directions to limit the interactions to just two devices (emitter and receiver). We've done it to clearly differentiate between the effects of wake impingement and standalone atmospheric turbulence; we didn't want to use wakes originating from the inside of the wind farm, as their shape could be distorted due to superposition effects. We agree that this could be potentially confusing to the reader if

not explained properly; we have added a necessary explanation to Section 2.2. of the revised manuscript, it can be found in lines 115-118.

13. Line 309: "Figure 6 illustrates the process for training the wind sensing model and demonstrates its post-training performance in producing wind field estimations." The application scheme of the model is shown here, but not post-training performance. Please adjust the sentence.

Agreed, we have revised the sentence as suggested.

14. Eq(1) & Eq(2) (and possibly others): don't use italic font for sine and cosine and subscript text (except variables).

Thank you for spotting this. We have corrected this in the revised manuscript.

15. Line 344ff: This is the first time that higher harmonics are mentioned. The Coleman transform was only described for the 0P / 1P harmonics. Also: Fig. 7 names an "original load", which suggests a (non-transformed) blade load. Meanwhile, it's said in line 344 that the rotor loads are decomposed into their frequency components. Is it correct, that you e.g. calculate the 3P share of a yaw moment? Or do you calculate the 3P of a blade load? A flow chart of the pre-processing steps would help.

Agreed, the original manuscript lacked clarity when describing the turbine response preprocessing steps. We have added a new block diagram (Fig. 5 in revised manuscript) that visualises the process. Also, we have modified the figure describing the frequency decomposition (Fig. 6 in revised manuscript), replacing the confusing 'Original load' with proper variables describing the rotor loads (yaw/ tilt moment etc.).

16. Line 351ff: Please elaborate on the temporal dependency and time lag. Which temporal scales of the turbine dynamics are you addressing here?

Our implementation of lagging is a simplified 'memory' functionality. With a simple implementation, it makes wind sensing less instantaneous and helps to capture the turbine response across several seconds. We have expanded on the paragraph.:

"In order to capture the short-term temporal dependencies and patterns in time series of all wind sensing inputs, the features are embedded with their lagged values. For each sample in a time series, two additional features expressing the past value of the curve are added. These lagged features are obtained by shifting the time series by 4 and 8 seconds from the current time stamp. This effectively makes the estimation more stable and noise-resistant, as the wind slice is reconstructed with turbine response across several seconds. Specific lag values used are determined by testing the framework's performance with a few different configurations and choosing the one that gives the best overall results."

17. Subsection 3.3.3: The DTC is a nice choice and the dimensionality reduction shown in Fig.8 looks appropriate. My only point regarding the wind field parametrization is: The here shown YZ-wind field slices are rectangular, while the rotor swept area is circular. The corners of the wind field thus include non-observable features, but could still influence the lower-order share of the DTC outputs. Were the plain rectangular slices used for the training? Or was any weighting or masking applied? Please comment on whether you expect an impact here.

We've used rectangular slices during the wind sensing training. At the current stage, there was no masking applied. You are absolutely correct that there are consequences of doing so – we have analysed the YZ-wise RMSE of wind sensing in the Results section (see Fig. 11 in revised manuscript). It is clear that the corners have the highest error due to blades being basically unable to clearly sense the flow fluctuations in these regions. For the revised manuscript, we included a discussion on that aspect in the Discussion section (Framework performance – Wind sensing subsection). We will consider applying masks in future work.

18. Section 3.3.4: Please add some more details and a literature source to the used regression approach.

To address your comment, we have expanded the description of our linear regression implementation (paragraph starting in line 215 of revised manuscript). Moreover, we have added a source describing the localised linear regression in more detail (Cleveland, 1988).

19. Line 392ff: Is the distinction of the four classes based on the constellation of the simulation run or based on the instantaneous wake position (which could differ due to the employed DWM model). Also: Please define the overlap margins, from which you categorize full/partial/no wake impingement.

The classes are defined based on the simulation setup – specific wind direction refers to specific class. The wind direction differs by 5 degrees between the fully and partially impinged cases, we have added this information to the revised manuscript. When the wake deficit falls between the CNN's understanding of partial and full impingement, the classification is less certain (e.g. 60% full, 40% partial). It is however not an issue for the overall performance, as shown in the specific wind fields we have analysed sample by sample. When the wake meanders from the centre to the side, the CNN changes its output appropriately (from full to partial impingement), and the two-dimensional Gaussian gets fitted nonetheless.

We are not entirely sure what you mean by 'overlap margins'. If you refer to quantifying the degree of misclassifications, we have added an appropriate confusion chart (Fig. 9 in revised manuscript) that shows which classes are the main source of error.

20. Section 3.5.1: The fitting function does not fully reflect the fit that was probably implemented. To fit a wind field with wake deficit, it should be U = u_ambient – f_G, here considering that parameter A is negative.

Agreed, the function from the original manuscript did not fully represent the fitting procedure. We have added an equation (Eq. 5 in revised manuscript) for the reversed wake deficit, which the bivariate Gaussian is actually fitted on.

21. Section 3.5.2: How does the low-pass filter deal with falsely identified no-impingement instances?

In current implementation, the moving average filtering is performed before the removal of samples identified as 'no wake impingement' from the characterised wake time series. This removal is implemented as assignment of NaN values to specific time stamps. This solution isn't perfect, as there are several gaps in time stamps where the wake can be assumed to be present. We have potential ideas on how this problem could be solved (Kalman filtering), which we discuss in the Discussion chapter. To make the reader aware of how the current implementation works, we have added a subsection 'Treatment of non-impinged samples' (indexed as 2.5.3 in revised manuscript):

To satisfy the assumption that wake characterisation should not be performed for wind samples without clear impingement, the filtered wake properties are treated for a given time stamp $i$ as follows:

$$y_c^{\text{filt}}(i), \sigma_y^{\text{filt}}(i) = \begin{cases} y_c^{\text{filt}}(i), \sigma_y^{\text{filt}}(i), & \text{if class}(i) = \text{fully/partially impinged} \\ \text{NaN}, & \text{if class}(i) = \text{no detectable impingement} \end{cases} \quad (9)$$

22. Line 456: Please rephrase this sentence.

We have rephrased the paragraph accordingly.

23. Fig. 11: Please add more details to the figure caption.

We have expanded the description: "Typical YZ distribution of RMSE in wind sensing (normalised by the corresponding Uamb value). Rotor outline marked with dotted white line."

24. Section 4.1: a diff-plot between estimated and simulated wind field would help to analyse, whether systematic or just random differences exist (e.g. the central deficit mentioned in line 500)

Fig. 11 in the revised the manuscript provides the information on wind sensing accuracy across the YZ plane. We believe that a diff-plot would be in this case redundant.

25. Line 501-503: "A slight disagreement would not be normally problematic; however, in this case where the flow has little overall turbulence, a subtle deficit like this can't 'hide' among other eddies, which could potentially result in classifying the

mentioned samples as being wake impinged." This sounds rather complicated and nested. Please rephrase the sentence.

We have reworded the entire paragraph due to modifying the Results section.

26. Fig. 14 shows that simulations for all wind directions are on hand. Please report this more explicitly in section 3.6.

We have reworded the section appropriately, now the testing setup is described more explicitly.

27. Fig. 14 @ 5 m/s: why is detection ratio different here? In partial load range, the non-dimensionalized wake should be similar, thus limited impact on the sensing is expected. Also: It would be good to know the turbine's cut-in wind speed. At 5 m/s undisturbed ambient wind speed, a wake-exposed turbine might experience a rotor-effective wind speed below cut-in.

There is a wind sensing anomaly that introduced a 'fake' wake to 5 m/s cases. We have now added additional metrics that quantitatively measure how often this anomaly occurs (confusion matrix in Fig. 9 of revised manuscript). The new Discussion section addresses this issue in more detail.

Unfortunately, due to SGRE's data protection, we are unable to include turbine details such as cut-in wind speed. We have nonetheless addressed this aspect in the Discussion/Applicability subsection, where we comment on the implications of poor performance in 5 m/s region.

*Discussion*

29. Line 591ff: ". Implementing other approaches such as Long short-term memory (LSTM) networks could potentially allow for the forecast to be extended to predict wake locations and meandering behaviour a few minutes ahead. Further research needs to be conducted to investigate these leads." Please provide a source here and a stronger supporting argument for the claim that the (stochastic)meandering wake location can be predicted by the receiving turbine. Otherwise, please consider softening this statement.

We may have indeed exaggerated the potential of using LSTMs here. We were unable to find proper sources supporting our statement of 'a few minutes ahead'; the literature generally suggests shorter time scales of forecast. With that in mind, we've softened the discussion and added sources:

" Relevant literature (Luo2024, Zhou2023) shows that Long Short-Term Memory (LSTM) networks could potentially provide a short-term forecast of the wake dynamics, thus providing an alternative solution. Further research needs to be conducted to investigate these leads."

*References*

30. Line 700: incomplete reference (journal, DOI)

Sorted.

**References**

Bottasso, C. L., Cacciola, S., & Schreiber, J. (2018). Local wind speed estimation, with application to wake impingement detection. Renewable Energy, 116, 155–168. https://doi.org/10.1016/j.renene.2017.09.044

Lio, W. H., Larsen, G. C., & Thorsen, G. R. (2021). Dynamic wake tracking using a cost-effective LiDAR and Kalman filtering: Design, simulation and full-scale validation. Renewable Energy,172, 1073–1086. https://doi.org/10.1016/j.renene.2021.03.081

Onnen, D., Larsen, G. C., Lio, W. H., Liew, J. Y., Kühn, M., & Petrović, V. (2022). Dynamic wake tracking based on wind turbine rotor loads and Kalman filtering. Journal of Physics: Conference Series, 2265(2), 022024. https://doi.org/10.1088/1742-6596/2265/2/022024

Petrovic, V., Jelavic, M., & Baotic, M. (2014). Reduction of wind turbine structural loads caused by rotor asymmetries. 2014 European Control Conference, ECC 2014, 1951–1956. https://doi.org/10.1109/ECC.2014.6862484

Schreiber, J., Bottasso, C. L., & Bertelè, M. (2020). Field testing of a local wind inflow estimator and wake detector. Wind Energy Science, 5(3), 867–884. https://doi.org/10.5194/wes-5-867-2020

Meyers, J., Bottasso, C., Dykes, K., Fleming, P., Gebraad, P., Giebel, G., Gocmen, T., and van Wingerden, J.-W. (2022). Wind farm flow control: prospects and challenges. Wind Energy Science, 7, 2271–2306. https://doi.org/10.5194/wes-7-2271-2022

Munters, W. and Meyers, J. (2018). Towards practical dynamic induction control of wind farms: analysis of optimally controlled wind-farm boundary layers and sinusoidal induction control of first-row turbines (2018). Wind Energy Science, 3, 409–425. https://doi.org/10.5194/wes-3-409-2018

Frederik, J. A., Doekemeijer, B. M., Mulders, S. P., and van Wingerden, J.-W. (2020). The helix approach: Using dynamic individual pitch control to enhance wake mixing in wind farms. Wind Energy, 23, 1739–1751. https://doi.org/10.1002/we.2513

Cleveland, W. S., Devlin, S. J., and Grosse, E. (1988). Regression by local fitting. Journal of Econometrics, 37, 87–114. https://doi.org/10.1016/0304-4076(88)90077-2

Luo, Z., Wang, L., Fu, Y., Xu, J., Yuan, J., and Tan, A. C. (2024). Wind turbine dynamic wake flow estimation (DWFE) from sparse data via reduced-order modeling-based

machine learning approach. Renewable Energy, 237, 121 552, https://doi.org/10.1016/j.renene.2024.121552

Zhou, L., Wen, J., Wang, Z., Deng, P., and Zhang, H. (2023). High-fidelity wind turbine wake velocity prediction by surrogate model based on d-POD and LSTM. Energy, 275, 127 525, https://doi.org/10.1016/j.energy.2023.127525

---

## Author Comment (AC2)

Thank you for taking the time to review our manuscript and for your valuable comments, which have helped us enhance the quality of the paper. Below, we include your comments in **black**, followed by our responses in **blue**.

**Review:** *Wind turbine wake detection and characterisation utilising blade loads and SCADA data: a generalised approach*

**General Comments**

This article presents a model for the detection and characterisation of wakes via estimating wind fields from blade load data. The model is well-explained, and the paper reads very clearly, with informative visualisations of the results. The approach developed here would be of interest to readers of this journal. There are a few areas that could be expanded on, including a deeper look into the accuracy of the wake impingement classification and wake characterisation. Additionally, a detailed flow-chart of the full process would guide others looking to reproduce these results.

Thank you, we appreciate the positive feedback. We agree that this methodology will hopefully be of interest to the readers and will contribute to the wake estimation field.

**Specific Comments**

1. Metrics / Accuracy:
   a) Is there a metric by which it is decided whether a wake is impinging on a rotor, e.g. a velocity deficit threshold? In line 39, "significant" impingement in terms of both magnitude and time is mentioned, but there is no further detail on the training data labelling for wake detection and classification. Could the details of how flow was classified as containing a wake be included in e.g. Section 3.4?

      For the purpose of this work, we did not specify such a metric, as there was no need to explicitly do so. It is largely due to the fact, that our wake detection model is based on classification by identifying patterns in the labelled data. The convolutional neural network learns the pattern of the wake deficit by being trained with an extensive sample library showing the four wake impingement conditions under varied wind speed and turbulence intensity values. Although such a 'black-box' approach is not based on physics of a wake deficit, it does offer high flexibility; with appropriate training data, it can easily capture various wake impingement conditions (e.g. multi-wake). Currently, the training data takes the 'wake impingement' samples from a single turbine at a fixed distance. Possibly, this will be expanded in the future. All in all, we agree that the original explanation of classes' definition could be improved, which is the section "Framework implementation: wake

detection" (2.4 in new numbering) was modified appropriately in the revised manuscript.

b) For the reported wake detection accuracy of 91% in Section 3.4, did this vary by "class" of impingement, e.g. was the model more or less accurate at predicting partial impingements? This may be relevant for future work using this model in wake steering controllers.

No, this is an overall metric for all 4 classes. We agree that this section could use a mention of how accurate prediction of each class is. Consequently, we added a confusion matrix (Fig. 9 in revised manuscript) that describes the results of integrated CNN testing (done automatically during the training procedure) in more detail. We believe this will allow the reader to immediately realise what are the strong and weak points of trained neural network, making their interpretation of results more informed.

c) It would be informative to include accuracy metrics of related wind flow estimators or wake classifiers, to provide context for the model(s) developed in this paper.

Although we do agree that this additional metric would be informative, we believe that adding a discussion subsection that would satisfy your comment is outside of the paper scope. The main novelty of this work is the introduction of a modular approach to the wake estimation problem – a generalised framework being a combination of several models. The models that we implemented can be easily swapped for something more accurate, and we want to encourage the community to use their methodologies this way.

Furthermore, a direct comparison with other detection/ characterisation studies would be difficult due to fundamentally different assumptions. Other works that consider the wake detection aspect, such as (Onnen et al., 2022) or (Bottasso et al., 2018), do not test their solutions under a full range of wind conditions as we did. Existing simulation-based wake characterisation studies such as (Onnen et al., 2022) use different approaches for establishing the reference.

Results:

a) When the trained model was tested on a new receiver turbine in Section 4, were metrics for the accuracy of the wake detection or classification models calculated? In particular for the wake impingement predictions under 9% turbulence intensity, were the results in e.g. Figure 15(d) confirmed to be sensible given the increase in turbulence compared to training data?

Excellent point, we agree that quantitative analysis would make this study better. We have spent significant time looking into the version of the DWM model that we're using, attempting to find the reference for wake detection

as per the other reviewer's suggestion. When computing the wind field at the receiving turbine, multiple meandering wake deficit profiles are in general imposed on an otherwise clean "Mann box", including additional added turbulence determined by the shape of the deficit profiles. To combine multiple overlapping wake deficit profiles, some rule is applied pointwise, that is, independency at each relevant grid position in the wind field. The rule we used was "pick the maximum deficit at the point", but other rules are possible. As a result, the wake deficit profile that is applied to the Mann box does not necessarily have a simple shape with a well-defined location of maximal deficit. For majority of wind directions where the flow is coming from the inside of the wind farm, there are always some influencing turbines registered by the code – even if they are too far away to have a clear effect on the 'receiver' device. As a result of the above, a reference such as 'waked/not-waked for a given wind direction' is impossible to establish directly from the simulations. This is partially dictated by the nature of our study - we have tested the detection performance for all wind directions, where the wake shedding turbines are at various upstream distances, hence the rate of wake breakdown/lateral displacement is also varied. Furthermore, to the best of author's knowledge, there isn't a widely-recognised definition of 'wake impingement' (e.g. by means of reduced power output) we could use here. As a work-around to this issue, we have provided a reference for the quantitative analysis in a 'synthetic' way. We've trained a new classifier analogically to the process described in Methodology, with the only difference being that the training dataset is derived from simulated wind fields, not the estimated ones. Without the bias from the wind field reconstruction, this classifier achieves approx. 99% accuracy under integrated testing, and its classifications are thus assumed to be precise enough to become 'ground truth' reference. Such a reference allows to consider the effects of varied wake dispersion under different ambient conditions, and arguably fits this analysis better than a general impingement definition based on inflow angle. All in all, the goal of our wake detector is to identify "clear wake impingement from a nearby device" as we define it in the article. The reference as seen in Fig. 12 (revised manuscript) is sensible, as the impingement ratio decreases with increasing wind speed – just what would happen in real life, as operating in the above rated region makes wake less pronounced. We consider the limitations and bias from this approach in the Discussion section.

b) Is there an explanation for the "fake" wakes seen in Figure 17(c) / 13(a), or a proposed method to alleviate this? These simulated areas were mentioned

as potentially resulting in mis-classification, is there any way to quantify how often this might occur?

Thank you for this comment, we agree that this aspect should be explained better. We believe that these 'fake' wakes originate from the training dataset selection. The wind sensing model was trained with simulations with a following distribution: 25% full impingement, 50% partial impingement, and 25% strictly ambient turbulence. By having one wind sensing estimator for all these cases, a bias is introduced – it is most likely a result of training the model with data where most cases show a wake. In the revised manuscript, we have introduced an additional discussion on that hypothesis (Discussion section, Framework performance/Wind sensing subsection). Moreover, we have added a confusion chart (Fig. 9 in the revised manuscript) that quantifies how many 'false positives' or 'false negatives' occur during the integrated testing of the classifier. This chart clearly shows that misclassification between 'full impingement' and 'no detectable impingement' is the main source of error, and gives a metric to how often this happens.

c)  Did the superposition of wakes or the position of the turbine deep within the farm have any effect on model's accuracy in Section 4?

Firstly, we did not see a particular effect of wake superposition. As seen in the polar wake detection plots, the wakes originating from the eastern turbines (a single machine upstream) did not yield substantially different wake detection results than the wakes originating from the direct NW or SW neighbour. It appears that the primary aspect that decides on the wake detection performance is the distance between the 'receiver' and 'emitter' machines. In the revised manuscript, we have introduced an additional discussion on that aspect (Discussion, Framework's performance subsection).

2. Flow Chart: It would be very useful to have a more detailed flow chart (i.e. more indepth version of Figure 3) that includes the steps take for e.g. pre-processing to extract wind field from turbine blade loads, fitting of DCT factors, constructing wind fields, sampling frequency and fitting wake parameters.

Although we do agree that it's generally good to provide the reader with a comprehensive diagram describing the methodology, we believe that there is a better alternative here. A diagram like this could potentially overwhelm the reader – the purpose of this section is to introduce a modular way of thinking to wake estimation problem, not introduce the specific implementation details. We aim for this framework to be reused by the research community, and they can

swap the models that we used for other methods that are leaner/better for their application. However, we do agree that a diagram that shows the rather complicated turbine response preprocessing would be very informative; we will discuss this further in your next comment.

3. Pre-Processing Diagram: A diagram of the turbine loads and how they are transformed would be informative in Section 3.3.

   As mentioned above, we agree that it would increase the readability, helping the reader understand the turbine response preprocessing. We have added an appropriate block diagram (Fig. 5 in the revised manuscript)

4. CNN Model: More detail on the CNN architecture is needed in Section 3.4.
   We have prepared an expanded description of a CNN (including examples, references, introducing its architecture components, and explaining what specific layers are responsible for) and put it in the Appendix A. In the beginning of the section, the readers are referred to it. The reason for this separation is brevity of the main article content. The scope of the paper is large, and the goal of our Methodology chapter is to provide the necessary implementation details and thus repeatability of this approach. The Appendix A serves as an introduction into the key concepts of CNN for readers unfamiliar with this technique. To address your comment, we have added a table (Table 3 in revised manuscript) describing the architecture in with the details necessary for implementation.

5. Conclusions: The conclusions are very brief, they should be expanded to include a summary of the accuracy of the models developed, as well as a short description of current limitations before future work.

   Excellent point, we have added a mention on the accuracy metrics of all models and addressed the current limitations.

**Technical Corrections**

1. General: Please ensure all acronyms are defined with capitalisation at the first use, and used consistently thereafter.

Excellent point, we have fixed the issue.

2. General: Please be consistent with using either double or single quote marks.

Sorted.

3. Line 25: Please clarify "the aforementioned task".

Sorted.

4. Line 27: "altered" does not give enough information about the features of waked flow that lead to higher loads, suggest re-wording to e.g. "experience a more turbulent wind field".

Sorted, thanks for the suggestion!

5. Line 34: "yaw control" usually refers to control of a single turbine to follow the inflow, the standard term for farm-wide yaw optimisation is "wake steering"; suggest using this term instead.

Sorted.

6. Line 50: "to date"

Sorted.

7. Line 52: Typo: "turbine's wake"

Sorted.

8. Line 65: Suggested re-word: "that the focus of the current work is to develop a solution"

Due to the feedback from the second review, we have already re-worded this sentence.

9. Line 87: Please include a reference for the microscale length scale.

For comments 9 – 22: General comment 2 from the second review suggested to transform the entire Background section into a shortened and more relevant literature review, and put it in the Introduction. We agree with this critique – in its original form, the Background mentioned a lot of concepts not being directly used in the rest of the work. As a result, this comment and several other below are referring to text which has been removed from the revised manuscript.

10. Figure 1: I think it should be $A1 = A0/(1 − a)$?

See comment 9.

11. Line 109: "as a wake"

See comment 9.

12. Line 110: Please include a reference for the "2-4 rotor diameters" statement.

See comment 9.

13. Line 119: Please include a reference for the Gaussian wake model relations.

See comment 9.

14. Line 125: I think "differs" is meant rather than "defers"?

15. Line 135: The explanation around atmospheric shear and the location of maximum turbulence needs more detail.

16. Line 145: For the infinite wind farm case, the power extraction from the turbines is balanced by entrainment of kinetic energy from the flow above; the explanation given here seems to reference increasing vertical height in the ABL?

17. Lines 153 & 154: Unclear wording, is the data from the first 10 rows and first 8 rows of turbines per farm? And is the power loss between 45% and 70%?

18. Line 181: Suggested re-word: "distinguishes the various impacts of turbulence"

19. Line 187: Suggested re-word: "The widely-discussed method introduced by..."

20. Line 196: Suggested re-word: "in incoming flow"

21. Line 217: The phrase "accurate approximate estimation" does not make sense.

22. Line 225: "wind farm flow control" for consistency.

23. Line 264: This sentence is quite convoluted, please re-word.

Ultimately, we've decided that this sentence is not only convoluted, but it also doesn't serve much purpose here. This information is being given at several other instances in the Section. For this reason, we have removed it completely.

24. Line 276: What kind of evaluation metrics were used to determine the model had reached sufficient accuracy?

In this work, we wanted to propose a wake characterisation model that would output not just the wake centre position (as is the usual approach in other wake tracking studies, see e.g. Onnen et al., 2022), but also a measure of the impinged area. The methodology that extracts these two time series can be proved very useful to closed

loop wind farm flow control schemes. Our original statement of 'sufficient accuracy' of this approach was not accurate, which is why we have reworded this sentence.

25. Line 298: Please specify whether "left" and "right" are as seen looking at the front or the back of the turbine.

Good point, added that information to the revised manuscript. The sentence now is: "Four wind directions, representing four distinctive wake impingement scenarios as seen from the front of the rotor, are defined as follows: (...)"

26. Line 314: A brief description of conditional dependence would be useful here.

We have added an enhanced explanation of how we apply the conditional dependence (lines 145-151 in revised manuscript). Readers interested in finding out more details are referred to the reference.

27. Line 315: Typo: "the following"

Due to the previous comment, the sentence was modified, so the typo is automatically resolved.

28. Line 348: Typo: "the the"

Sorted.

29. Line 354: "with a few"

Sorted.

30. Line 358: Could a brief list of all the inputs be given, either in the text or as a table, for clarity on what the 96 variables are?

We have added a table summary of the extracted features (Table 2 in revised manuscript).

31. Line 361: Suggested re-word: "are being processed"

Not exactly sure what do you refer to – in the original manuscript, the wording is "are being processed". Changed it to "are processed" for sharpness.

32. Line 382: More description of the "simple models" is needed.

To address your comment, we have expanded the description of our linear regression implementation. Moreover, we have added a source describing the localised linear regression in more detail. These can be found at lines 215-226 of revised manuscript.

33. Lines 396 & 411: It would be easier to read the proportions than the actual numbers of simulations, e.g. 10% instead of 1,120 on line 411.

Sorted, we have replaced 1,120 with 10% (line 258 in revised manuscript)

34. Line 414: Suggested re-word: "case was approximately 91%"

Sorted.

35. Line 430: Definition of the "rotation angle" needed.

Reworded the sentence to "$\rho$ can be used to calculate the lengths of semi-minor and semi-major axis of the wake ellipse, as well as their orientation with respect to the YZ axes."

36. Line 433: "D" has already been used as dimension e.g. "2D".

To avoid confusion, in the context of dimensionality, we have replaced all instances of D with "-dimensional". So for example, the "2D Gaussian fit" is now referred as "two-dimensional Gaussian fit".

37. Equation 6 (Line 446): Is the integral missing $dt$?

Thank you for spotting this, sorted.

38. Section 3.6: This would make more sense as the first part of Section 4.

Sorted, moved this to the first subsection of Results section in the revised manuscript.

39. Line 460: Suggested added wording: "centrally located within the wind farm"

Sorted.

40. Line 480: "have a low mean RMSE"

Sorted.

41. Line 503: Suggested re-word: "simulations showed that"

Sorted.

42. Line 505: The term "amount of turbulence" is ambiguous, since the turbulence intensity has not changed but the wind speed has increased. Please re-word for a clearer explanation.

Due to the other comments, the Results section has been modified significantly. The sentence is no longer present in the manuscript.

43. Line 527: Suggest using "South and East" rather than "S and E".

Sorted.

44. Line 534: Typo: "inverse"

Sorted.

45. Line 598: Suggested re-word: "after the consideration".

Sorted.

46. Line 599: Suggested re-word: "Firstly, the wind farm flow control brings the largest benefits for below rated operation"

Sorted.

47. Line 620: Suggested re-word: "(and solution to some of"

Sorted.

48. Line 643: "2D" and "1D" without hyphen for consistency.

Sorted.

**References**

Bottasso, C. L., Cacciola, S., & Schreiber, J. (2018). Local wind speed estimation, with application to wake impingement detection. Renewable Energy, 116, 155–168. https://doi.org/10.1016/j.renene.2017.09.044

Onnen, D., Larsen, G. C., Lio, W. H., Liew, J. Y., Kühn, M., & Petrović, V. (2022). Dynamic wake tracking based on wind turbine rotor loads and Kalman filtering. Journal of Physics: Conference Series, 2265(2), 022024. https://doi.org/10.1088/1742-6596/2265/2/022024

---

## Author Comment (AC3)

**WES-2025-17: Wind turbine wake detection and characterisation utilising blade loads and SCADA data: a generalised approach**

**Response to Editor**

Dear Cristina,

Thank you for handling the review process for this paper. We are grateful to the reviewers for providing excellent suggestions and feedback which will help improve this manuscript.

The suggestions that majorly impact the structure of the manuscript are:

a) RC2, General comment 2: the reviewer advised to sharpen the literature review from the initial Background section, removing sections not directly relevant to the study. As a result, specific subsections (containing relevant state-of-the-art relevant to the research gap definition), are now used in Introduction, and the rest of the Background section is removed.

b) RC2, General comment 7 / RC1, Specific comment "Results a)": both reviewers have suggested to add a quantitative analysis for wake detection and characterisation. We have done so, and the revised manuscript includes additional metrics and figures that allow for precise comparison of the models' accuracy across different wind conditions. Due to this fundamental change in approach and not relying mostly on qualitative/visual analysis, the Results section has been significantly reshaped.

c) RC2, Comment 6: the reviewer advised to revise the Results section and move the appropriate discussions to the Discussion section. Consequently, the new Results section is more concise, and Discussion now contains an extended commentary of framework performance.

Other suggestions mostly include requests for additional clarifications, plots and mistake corrections. We look forward to submitting a revised version of the manuscript. If anything in our responses requires further clarification, we'll be happy to provide it.

Piotr Fojcik (on behalf of all co-authors)

---

## Referee Report (RR1)

**Review: *Wind turbine wake detection and characterisation utilising blade loads and SCADA data: a generalised approach: Revision 1***

**General Comments**

The authors have re-structured their manuscript, resulting in a more concise paper that is easy to follow and reads well. They have addressed the majority of comments raised, and there are only a few further points I recommend addressing as listed below.

**Specific Comments**

1. Wake classifications: It would be useful to have a clear statement on how the training (and testing) datasets for different wake impingement categories were labelled. There are occasional references to different wind directions (5 degrees on line 122) or "manual review" on line 238 – if the wake impingements for training were all classified manually / by eye, then this should be stated.
2. Section 2.4: The removal of all ambiguous training data (~line 242) invites the question of how the final model would classify such cases. Given the model outputs probabilities of different impingement classes, it should be able to handle a wake that is a "edge case" between e.g. fully and right-impinged by predicting ~50% probability of both. Some additional text around this could help to clarify why training on these data would be an issue as presumably they could be given labels of equal probability between the two potential classes?
3. Figs 16 & 17: Compared to the previous version of the manuscript, Figures 16 and 17 are paired with different $U_{amb}$ values in the current version. Switching these wind fields / speed values would also make more sense with the discussions on these figures in Section 4.2.

**Technical Corrections**

1. Abstract: I recommend not using acronyms in the abstract as they have not been defined; however RMSE and DWM are probably known to readers of this journal.
2. Lines 118, 321, 404: Suggest writing out "approximately" rather than "approx."
3. Lines 196, 204, 231: Italicise "*U*" for consistency with first use on line 154.
4. Line 199: Capitalise "Discrete Cosine Transform" in DCT acronym definition.
5. Line 248: Suggest "monotonically" rather than "iteratively", or delete the word "iteratively".
6. Equation 7: Given "t" is part of the limits, it shouldn't be a variable in the integral – suggest changing the instances of "t" within the integral to something else e.g. $\theta$
7. Line 310: Suggest re-wording title of Section 3.1 to e.g. "Performance evaluation"
8. Table 5: For ease of comparison, could the two types of RMSE be presented in the same way e.g. percentages for both?
9. Equation 11: Variable "i" has been used in previous equations to represent timestep, please use a different variable for wind direction.
10. Lines 380, 453: It reads as though full impingements are under (high or) low $I_{amb}$ and partial are under the other $I_{amb}$.
11. Line 386: Suggest removing "raw" before simulated, or re-wording to e.g. "simulated (rather than estimated)"
12. Line 450: Suggested re-word: "with **a** few gaps"
13. Line 465: Suggest removing "the" from "after the consideration"

14. Line 471: Suggested re-word: "Wake steering control brings **the** largest benefits"
15. Line 483: Suggested re-word: "post-processing analysis which account**s** for"
16. Line 493: Is the acronym "LSTM" ever used? If not, no need to define.
17. Line 507: The sentence beginning "Wake detection" could be worded more clearly, and context given to the RMSE e.g. writing as a % of times the correct type of impingement is identified.
18. Line 515: Suggested re-word: "accounting **for** more"

---

## Referee Report (RR2)

**Paper Title:**

*Wind turbine wake detection and characterisation utilising blade loads and SCADA data: a generalised approach*

Dear Authors,

Thank you for addressing the comments such detailed. I appreciate the implementations you made, and I especially like the increased level of discussion. Below, I gather a short list of additional comments on the revised manuscript. The line numbers I state refer to the document with tracked changes.

**Comments:**

1. Line 61: "Despite achieving great performance […]" please formulate this more neutral. Same as the load-based methods, the lidar-based concepts are not perfect. Their drawbacks are often a compromise between spatial or temporal observability (depending on staring / scanning lidar).

2. Line 70: Onnen et al. (2022) do not make use of the in-plane blade loads, only out-of-plane.

3. Line 80: "The up-to-date wake detection studies analysed the wake impingement in a scenario with a single upwind turbine […]" – This is certainly a valid point. But does this paper fill this gap? In your author's response you mention that the wake overlap method in the simulation environment is "pick the maximum deficit at the point". So is the method really tested for complex overlapping wakes? Meanwhile you have a strong point showcasing the method for wakes shed at various upstream distances.

4. Regarding comment 19 from the first review round: By 'overlap margins' I mean the definition of 'full' or 'partial' wake. E.g. a wake position y<0.25 D with respect to the wake exposed turbine might be denoted full wake, a position 0.25D < y < … denoted partial wake. I think your answer partly addresses this point already but I cannot see it in the manuscript. If it's not possible for you to state this, since the training data only considers inflow angles, please mention this in the manuscript. At the moment it says "The wind direction differs by 5 degrees between the fully and partially impinged cases. This setup allows to clearly differentiate between the effects of full/partial wake impingement and standalone atmospheric turbulence."
   The confusion chart is a good idea and helped here!

5. Nice that you added explanation to the unexpected performance at 5 m/s and the role of training data here.

---

## Author Response (AR2)

Thank you to both reviewers for taking the time to review our restructured manuscript and for your valuable comments. Below, we include your comments in **black**, followed by our responses in **blue**.

Considering that this round of revisions will result in a third manuscript iteration, in our response we will term the versions as follows to avoid confusion:

- Manuscript A: original submission from 12 Feb 2025

- Manuscript B: first revision submitted on 14 Apr 2025

- Manuscript C: second revision submitted on 01 Jun 2025

**Report 1**

Review: *Wind turbine wake detection and characterisation utilising blade loads and SCADA data: a generalised approach: Revision 1*

**General Comments**

The authors have re-structured their manuscript, resulting in a more concise paper that is easy to follow and reads well. They have addressed the majority of comments raised, and there are only a few further points I recommend addressing as listed below.

We appreciate your kind feedback. We are happy to hear that the changes we implemented addressed your concerns.

**Specific Comments**

1. Wake classifications: It would be useful to have a clear statement on how the training (and testing) datasets for different wake impingement categories were labelled. There are occasional references to different wind directions (5 degrees on line 122) or "manual review" on line 238 – if the wake impingements for training were all classified manually by eye, then this should be stated.

The samples representing each class are taken only from a specific simulation subset that shows a given wind direction. Manual inspection refers to the action of us making sure that the samples are showing what they are supposed to. Thanks to that action, we were able to realise that wind speeds over the 15 m/s threshold could not serve as the training data, under the assumptions that we made that all samples need to portray a clear wake. We have modified the entire section 2.4 (now titled Framework implementation: wake detection) to address this and the next comment. The labelling process is now more explicitly explained.

2. Section 2.4: The removal of all ambiguous training data (~line 242) invites the question of how the final model would classify such cases. Given the model outputs probabilities of different impingement classes, it should be able to handle a wake that is a "edge case" between e.g. fully and right-impinged by predicting ~50% probability of both. Some additional text around this could help to clarify why training on these data would be an issue as presumably they could be given labels of equal probability between the two potential classes?

The data from the 15-25 m/s range is eliminated, because in the simulations defined as full or partial wake impingement (which would be used for the full/partial labelled samples, as explained above) a clear wake deficit was often missing from the YZ snapshot samples. We believe that is due to a) higher meandering that these wind fields exhibit, b) weaker wake deficit resulting from smaller thrust force in the above-rated operation. Consequently, we would feed the training process with samples that are labelled as 'wake impingement' even when they do not portray a wake. To avoid a tedious process of manual labelling in this ambiguous range, and because we already had an abundance of training data from the 5-15 m/s range, we decided to eliminate the problematic dataset. Ultimately, the wake steering brings largest benefits in the low-wind conditions, so this action actually tailors the training data to the specific application. We have modified the entire section 2.4 (now titled Framework implementation: wake detection) to address this and the previous comment. We have added our reasoning behind the elimination of ambiguous samples.

3. Figs 16 & 17: Compared to the previous version of the manuscript, Figures 16 and 17 are paired with different Uamb values in the current version. Switching these wind fields speed values would also make more sense with the discussions on these figures in Section 4.2.

The wind fields used for the detailed analysis shown in Fig. 16 and 17 (manuscript B/C) have been changed compared to the original (manuscript A) for several reasons. Firstly, the new quantitative analysis allowed to properly look into the models' accuracy and describe it across several wind conditions (Table 5 of manuscript B). As a result, we believe that a detailed qualitative analysis of six different wind fields is no longer necessary, as the discussion can now strongly rely on the more representative quantitative results. For this reason, we have trimmed the selection to 4 wind fields. Secondly, the entire structure of the Results section has changed between manuscripts A and B; the visual inspection of wind sensing, wake detection and wake characterisation is now showed on a single figure for each wind field. This was done to satisfy the reviewers' comments and make the paper more concise, and to make the narrative flow better in light of the new quantitative results. We believe that showing six different wind fields in the expanded manner (4 subplots per wind field) could overwhelm the reader, especially considering the manuscript is already quite long. With

that in mind, and to avoid redundancy, on Fig. 14-17 we are showing just the four wind fields portraying four significantly different cases (wake impingement under high and low turbulence intensity, no impingement under high and low wind speed). We believe that such visual representation of four distinct wind fields is enough to inform the reader about performance in various conditions and provides a lot of content to assist the discussion.

**Technical Corrections**

1. Abstract: I recommend not using acronyms in the abstract as they have not been defined; however RMSE and DWM are probably known to readers of this journal.

We have removed SCADA and DWM acronyms. We believe that RMSE is such a wide-used technique in the field, that it doesn't require explanation.

2. Lines 118, 321, 404: Suggest writing out "approximately" rather than "approx."

Sorted.

3. Lines 196, 204, 231: Italicise "U" for consistency with first use on line 154.

Sorted.

4. Line 199: Capitalise "Discrete Cosine Transform" in DCT acronym definition.

Sorted.

5. Line 248: Suggest "monotonically" rather than "iteratively", or delete the word "iteratively".

We have removed the word "iteratively".

6. Equation 7: Given "t" is part of the limits, it shouldn't be a variable in the integral – suggest changing the instances of "t" within the integral to something else e.g. θ

We have changed the instances of "t" with theta as you suggested.

7. Line 310: Suggest re-wording title of Section 3.1 to e.g. "Performance evaluation"

Sorted.

8. Table 5: For ease of comparison, could the two types of RMSE be presented in the same way e.g. percentages for both?

Good point – we have changed the RMSE_det, it is now in percentage instead of fraction of 1.

9. Equation 11: Variable "i" has been used in previous equations to represent timestep, please use a different variable for wind direction.

Good point – we have replaced "i" with "k".

10. Lines 380, 453: It reads as though full impingements are under (high or) low Iamb and partial are under the other Iamb.

We have reworded to: "Table 6 shows the wake characterisation RMSE calculated for four example wind fields, comparing the model's accuracy for full and partial wake impingement under high and low I_amb"

11. Line 386: Suggest removing "raw" before simulated, or re-wording to e.g. "simulated (rather than estimated)"

Thank you for your suggestion – we have reworded to "simulated (rather than estimated)".

12. Line 450: Suggested re-word: "with a few gaps"

Sorted.

13. Line 465: Suggest removing "the" from "after the consideration"

Sorted.

14. Line 471: Suggested re-word: "Wake steering control brings the largest benefits"

Sorted.

15. Line 483: Suggested re-word: "post-processing analysis which accounts for"

Sorted.

16. Line 493: Is the acronym "LSTM" ever used? If not, no need to define.

Sorted.

17. Line 507: The sentence beginning "Wake detection" could be worded more clearly, and context given to the RMSE e.g. writing as a % of times the correct type of impingement is identified.

We have reworded to: "Wake detection model correctly responds to the wake presence; averaging the RMSE across all wind fields used in testing indicates that the correct wake impingement case is identified for approximately 77% of the samples."

18. Line 515: Suggested re-word: "accounting for more"

Sorted.

**Report 2**

**Revision 2: Wind turbine wake detection and characterisation utilising blade loads and SCADA data: a generalised approach.**

Dear Authors, Thank you for addressing the comments such detailed. I appreciate the implementations you made, and I especially like the increased level of discussion. Below, I gather a short list of additional comments on the revised manuscript. The line numbers I state refer to the document with tracked changes.

Thank you for your feedback. We are happy to hear you liked the extended discussion.

**Comments**

1. Line 61: "Despite achieving great performance [...]" please formulate this more neutral. Same as the load-based methods, the lidar-based concepts are not perfect. Their drawbacks are often a compromise between spatial or temporal observability (depending on staring / scanning lidar).

Good point. We have reworded to "Despite achieving good performance [...]". We believe that this wording describes their performance in a short but representative way.

2. Line 70: Onnen et al. (2022) do not make use of the in-plane blade loads, only out-of-plane.

Good catch, thank you. We have corrected the manuscript accordingly.

3. Line 80: "The up-to-date wake detection studies analysed the wake impingement in a scenario with a single upwind turbine [...]" – This is certainly a valid point. But does this paper fill this gap? In your author's response you mention that the wake overlap method in the simulation environment is "pick the maximum deficit at the point". So is the method really tested for complex overlapping wakes? Meanwhile you have a strong point showcasing the method for wakes shed at various upstream distances.

Thank you for this comment, we do agree that the research gap and how we fulfil it needs to be clearly defined. To clarify: in our wind farm simulation approach, every point at the 'wake receiver' (front-view) rotor plane, calculates the local wind speed independently. For each point on the grid, the strongest influencing upstream turbine is selected, and the corresponding shedding wake deficit is used to calculate the wind speed value. As a result, while we do not directly superpose the wake deficits at each rotor point, we are able to capture two wake deficits on the wake-receiving rotor plane; in other words, we could see one very wide deficit that is the combination of two wake deficits. This is, of course, a simplified model of wake overlap – but it gives reasonable approximation when it comes to e.g. power outputs of a turbine under multiple wakes. We didn't see such approaches in other wake detection/characterisation studies, which

is why we believe it is reasonable to include this information in our statement of research gap. To satisfy your comment, we have decided the modify the text as follows:

- Statement of research gap, the fixed wake shedding distance is now strongly mentioned (line 70, manuscript C): "The up-to-date wake detection studies analysed the wake impingement in a scenario with a single upwind turbine at a fixed distance, for a fixed set of wind directions."

- Contribution to the research gap, we specify that our key contribution is testing the wake estimation under various shedding distances (line 84, manuscript C): "The trained models are tested under a full range of wind directions within a virtual offshore wind farm, evaluating the method's performance for wakes shed at various upstream distances."

- Discussion on the effect of wake superposition to wake detection performance, we now explicitly mention how the wake deficits are combined – and this is the primary aspect why we don't see the effect of superposition (line 436, manuscript C): "This is likely a consequence of the wake superposition approach used, where wind speed at each YZ grid point on the receiving turbine is derived based on the strongest influencing wake deficit."

4. Regarding comment 19 from the first review round: By 'overlap margins' I mean the definition of 'full' or 'partial' wake. E.g. a wake position y<0.25 D with respect to the wake exposed turbine might be denoted full wake, a position 0.25D < y < … denoted partial wake. I think your answer partly addresses this point already but I cannot see it in the manuscript. If it's not possible for you to state this, since the training data only considers inflow angles, please mention this in the manuscript. At the moment it says "The wind direction differs by 5 degrees between the fully and partially impinged cases. This setup allows to clearly differentiate between the effects of full/partial wake impingement and standalone atmospheric turbulence." The confusion chart is a good idea and helped here!

Thank you for clarification. In our work, we have made the distinction between full and partial wake simply by changing the inflow angle. The wake (as seen in the front-view YZ snapshots) differs between these setups, which provides the basis for three specific partial impingement classes that we then use in the CNN-powered image recognition. The overlap margins that you mention seems like a good and robust idea, and in future we might try the wake detection this way. It is worth noting, that even using our simple class definition, the wake detector is able to switch between full and partial impingement reasonably well (referring to the probabilities that the CNN outputs, Figures 14-17 in manuscript B/C). We have modified the entire section 2.4 (titled Framework implementation: wake detection) to address your comment and make the labelling process more clear.

5. Nice that you added explanation to the unexpected performance at 5 m/s and the role of training data here.

Thank you, we are happy to hear you liked that aspect.